# Speed modulations in grid cell information geometry

**Zeyuan Ye** ✉ **& Ralf Wessel**

Grid cells, with hexagonal spatial firing patterns, are thought critical to the brain's spatial representation. High-speed movement challenges accurate localization as self-location constantly changes. Previous studies of speed modulation focus on individual grid cells, yet population-level noise covariance can significantly impact information coding. Here, we introduce a Gaussian Process with Kernel Regression (GKR) method to study neural population representation geometry. We show that increased running speed dilates the grid cell toroidal-like representational manifold and elevates noise strength, and together they yield higher Fisher information at faster speeds, suggesting improved spatial decoding accuracy. Moreover, we show that noise correlations impair information encoding by projecting excess noise onto the manifold. Overall, our results demonstrate that grid cell spatial coding improves with speed, and GKR provides an intuitive tool for characterizing neural population codes.

In navigation, it is crucial that the brain forms a certain internal representation of the external space[1]. Grid cells are widely regarded as an essential component of internal spatial representation[2,3]. Their hexagonal spatial firing patterns are thought to form a coordinate system of external space[4] and support the downstream hippocampal spatial representations (e.g., place cells)[5–9]. Yet maintaining precise spatial coding is particularly challenging at high running speeds, when self-location changes rapidly[10]. The effect of running speed modulation on grid cell population coding remains unclear.

Previous literature offers dual possible predictions about running speed modulation on grid cell codes. On the one hand, speed may support grid cell spatial coding. Running speed is known to mostly increase grid cell firing rates[11–13]. Rats running at high speeds (10 cm/s to 50 cm/s) are also known to have more medial entorhinal cortex (MEC) cells coding spatial information than when running at a low speeds (2 cm/s to 10 cm/s)[10]. On the other hand, speed signals disrupt the phase differences between pairs of grid cells[14]. Increasing speed may also lead to larger input noise (possibly from the medial septum[11,15] or speed cells[16,17]), causing larger noise error to accumulate over time, thus degrading spatial coding fidelity[18–21].

While these previous studies provide insights into speed modulations of grid cells, their analyses were limited to individual or pairs of grid cells[11–14] (although decoding analysis has been performed on

MEC cell population[10]). Neurons in the brain represent information through their collective population activity. Population noise covariance can significantly impact information coding[22–28]. Grid cells' activities are especially known to be tightly coupled and change coherently[14,29]. To study the speed modulation of grid cell code, it is important to analyze simultaneously recorded grid cell population activities, including the effect of noise covariance. Yet such a study is still lacking.

Because neural data is intrinsically high-dimensional, inferring the noise-covariance matrix can be difficult. A standard approach for discretely valued information is to compute the sample covariance across trial responses[24]. To handle continuously varying stimuli (e.g., orientations of static grating stimuli), one typically first bins continuous parameters, then collected repeated trials at each bin[30–33]. From these trial-based data, the noise covariance can be estimated using the sample-covariance estimator or, more recently, via a Wishart-process model[34].

However, discretizing continuous stimuli and collecting trial-based data can be impractical for two main reasons. First, high-dimensional inputs—such as natural images—require an exponentially large set of discretized values[27]. Second, many naturalistic experimental paradigms (e.g., navigation tasks) lack repeated trials[31,33,35]. A study on retinal representations of natural images addressed these

Department of Physics, Washington University in St. Louis, St. Louis, MO, USA. ✉e-mail: y.zeyuan@wustl.edu

challenges by substituting retinal data with convolutional neural network (CNN) units, explicitly formulating the noise covariance[27]. However, this approach relies on the observed similarities between retinal neurons and CNN units[36]. There's a trend in neuroscience to move beyond trial-based experiments, towards trial-free naturalistic experiments[31,33]. Yet, to our knowledge, it remains challenging to reliably estimate noise covariance from high-dimensional neural data in naturalistic tasks without repeated trials.

In this paper, we introduce Gaussian Process with Kernel Regression (GKR), a method for inferring both the smooth mean (manifold) and noise covariance from high-dimensional neural data, including recordings from naturalistic tasks. The study of manifolds and noise covariance falls within the framework of information geometry[27,37]. We applied GKR to simultaneously recorded grid cell activities[35]. We found that: (1) Running speed both dilates the grid cells' toroidal-like manifold and increases noise; (2) Nevertheless, the effect of manifold dilation outpaces the effect of noise increase, as indicated by the overall higher Fisher information at increasing speeds, and further supported by improved spatial coding accuracy at higher speeds; (3) Furthermore, compared to hypothetical independently firing grid cells, we found that noise correlations in real grid cells "shape" the noise structure such that more noise is projected onto the manifold surface, indicating that noise correlation in grid cells is information-detrimental. Overall, our results indicate that running speed enhances grid cell spatial coding through geometric modulations. GKR provides a useful tool to interpret noisy neural data from an intuitive information geometry perspective.

## Results

### Grid cell population spatial coding accuracy improves with increasing speed

We analyzed grid cell recordings from Gardner et al.[35], obtained as rats foraged in an open-field (OF). The dataset included approximately 60–200 simultaneously recorded grid cells per module, with the exact number varying by rat, recording day, and module (see Methods). The experiment comprised nine configurations, denoted using a notation system, for example, "R1M2" refers to rat R on day 1, specifically from grid cell module 2. Grid cells within the same module shared a similar

spatial period but differed in phase. Raw spiking data were converted into firing rates using a kernel averaging (see Methods), with example rate maps shown in Fig. 1A and Supplementary Fig. 1.

We analyzed the rats' behavioral data and found a predominance of low-speed movement, resulting in a concentration of observations in the slow-speed range (Supplementary Fig. 2). Since the primary goal of this study is to compare grid cell representation properties across different speeds, it is essential to control nuisance factors—specifically, ensuring an equal number of data points across speed conditions. To achieve this, we randomly sampled data within discrete speed bins (bin width = 5 cm/s, see Methods) to create a balanced dataset, $D_s$, with an equal number of data points per bin. We generated fifty such $\mathscr{D}_s$ datasets per experimental configuration to estimate uncertainty.

One straightforward approach to evaluate the quality of neural spatial coding is by decoding location information from neural states. Good decoding performance indicates good neural spatial coding. We designed a locally linear classification accuracy to evaluate the quality of spatial coding, formally referred to as spatial coding accuracy (SCA) (Fig. 1B, see Methods). Specifically, at each speed bin value, several locations were randomly sampled. For each sampled location, we created two conjugate boxes centered near the sampled location but positioned in opposite directions, separated by a fixed distance of 10 cm. Data from these two boxes were collected, relabeled as class 1 and class 2, and then split into training and test sets. A logistic regression model was then trained to classify the data and evaluated on the test set. The classification accuracy, averaged over all randomly sampled spatial locations, is referred to as the SCA. SCA quantifies how well neural states corresponding to two nearby spatial locations are discriminable.

For each sampled dataset $\mathscr{D}_s$, we computed SCA across eight speed bins as described above. Fifty $\mathscr{D}_s$—each yielding eight metric-speed pairs (where metric is SCA in this case)—produced a metric-speed dataset of 400 points (dots in Fig. 1C). A natural idea is to fit a simple linear regression (e.g., least-squares regression) to these 400 points and use the slope to quantify how metric varies with speed. However, this approach is problematic as it assumes all observations are independent, which here is violated: our 400 points come from fifty $\mathscr{D}_s$ drawn from the same original dataset $D$, so they are statistically

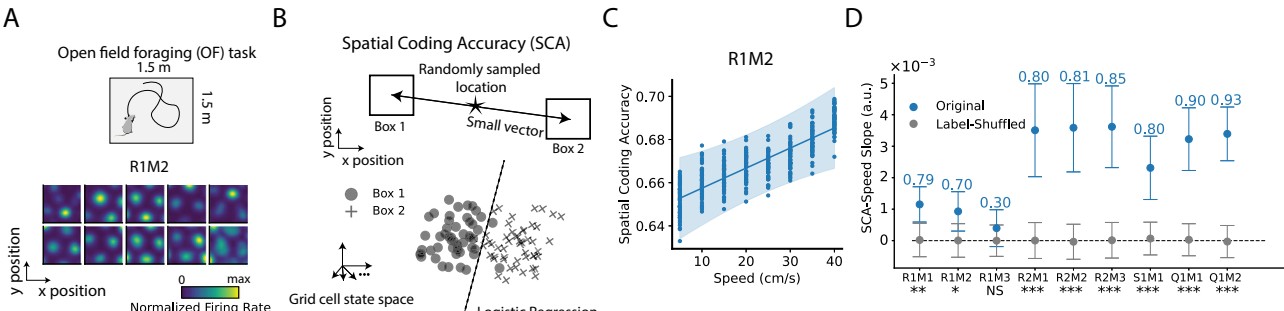

**Fig. 1 | Grid cell population spatial coding accuracy (SCA) improves with increasing speed. A** Top: Rats were performing open-field foraging (OF) tasks with grid cell activity collected (data from Gardner et al.[35]). Nine experimental configurations were analyzed, varying across rats, recording days, and grid cell modules. For example, the configuration "R1M2" represents rat "R", day 1, and grid cell module 2. Each experimental configuration was subsampled, generating 50 sampled datasets $\mathscr{D}_s$, ensuring a similar number of data points across different speed values (see text and Methods). Bottom: Example rate maps of grid cells in R1M2; with additional examples provided in Supplementary Fig. 1. **B** SCA quantifies the fidelity of grid cell spatial coding. Random locations were sampled within the open field, and for each location (star), two conjugate boxes were defined, separated by a fixed distance of 10 cm. Neural activity within these boxes was collected and classified using logistic regression (circles and crosses). The SCA is defined as the

average classification accuracy across all sampled locations (see Methods). **C** SCA as a function of the rat's speed. Each dot represents the SCA computed from a sampled dataset $\mathscr{D}_s$ at one speed bin (see Methods, fifty sampled $\mathscr{D}_s$ with eight speed bins = four hundred data points). The solid line and error band show the best-fitting line and 95% confidence interval (CI), estimated using Bayesian linear ensemble averaging (BLEA, see text and Methods). **D** SCA-speed slopes of different experimental configurations. Dots and error bars represent mean and 95% CI of the slopes estimated using BLEA (panel C shows the case of R1M2). Numbers above error bars are linear models' r-squared values. Asterisks below the x-axis denote the statistical significance of the slope computed from the original datasets $\mathscr{D}_s$ compared to label-shuffled data (two-sided, Bayesian method, see Methods). ***$p < 0.001$; **$p < 0.01$; *$p < 0.05$; NS no significance. Source data are provided as a Source Data file.

related. If one were to increase the number of sampled datasets $\mathscr{D}_s$, a naïve linear regression would misleadingly drive the slope's estimation uncertainty toward zero.

To address this, we introduced Bayesian Linear Ensemble Averaging (BLEA, see Methods). BLEA proceeds in two stages: first, it fits a Bayesian linear regression separately to the metric-speed data from each $\mathscr{D}_s$, yielding fifty posterior distributions over the regression weights; then it combines these posteriors via Bayesian model averaging[38,39]. The result is a Gaussian-approximated posterior over the slope (and intercept) together with predictive distributions, from which confidence intervals (CIs), p-values, and other statistical measures can be estimated using a Bayesian framework (see Methods)[40].

Applying BLEA to SCA (Fig. 1C, D), we found that SCA increases with speed, with a slope significantly larger than that of the label-shuffled dataset. This result holds across different classifiers used for computing the SCA (support vector machine and perceptron, see Supplementary Fig. 3), indicating better self-location representations in grid cell population at higher speeds.

## Gaussian Process with Kernel Regression (GKR) method for fitting manifold and covariance matrices from noisy neural states

What are the underlying neural mechanisms contributing to the improved spatial coding in grid cells? To explore this question, we need a tool to analyze the high-dimensional noisy neural states. A recent popular neural population geometry framework suggests that, instead of analyzing noisy high-dimensional data, it will be more intuitive to use certain methods to extract the data's underlying smooth manifold along with the noise covariance[34,41,42]. Wishart process is such a method that can infer the smooth manifold and covariance matrix[34]. However, the recent implementation of Wishart process requires trial-based experimental paradigm, which forbids this method to be used in broader and complex natural behaving experiments[34]. The OF task (Fig. 1A) is one of such natural behaving experiments without strict repeated trials.

Therefore, we developed Gaussian Process with Kernel Regression (GKR) method. The main purpose of GKR is to estimate the representation manifold relevant to the (known) labels of interest[42]. Response fluctuations arising from nuisance latent variables (e.g., emotional states) are captured as a noise covariance term. For example, in this paper, locations and speeds are the labels of interest, while neural response fluctuations due to other factors are treated as "noise," summarized in the noise covariance matrix.

Here we illustrate the principles of GKR. A dataset (e.g., $\mathscr{D}_s$) contains noisy neural states $\mathbf{r}$ whose dimensionality equals the number of neurons; and labels $\mathbf{x}$ whose dimensionality equals the number of label variables. A label variable can be stimulus parameters (e.g., grating stimulus' orientation), latent variables (e.g., internal decision factor) or behavior variables (e.g., x, y locations and speed). GKR assumes that $\mathbf{r}$ follows a Gaussian distribution

$$\mathbf{r}(\mathbf{x}) = \boldsymbol{\mu}(\mathbf{x}) + \mathcal{N}(\boldsymbol{\epsilon}; 0, \Sigma(\mathbf{x})) \qquad (1)$$

where $\boldsymbol{\mu}(\mathbf{x})$ is assumed to be a smooth-varying mean, also called a manifold in this paper; $\Sigma(\mathbf{x})$ is a smooth-varying covariance. The combination of manifold and noise covariance is referred to as a statistical manifold of neural response. The goal of GKR is to infer the manifold and covariance from a dataset $\{\mathbf{r}, \mathbf{x}\}$.

GKR solves this inference problem by two steps (Fig. 2A, see Methods): (1) inferring smooth manifold $\boldsymbol{\mu}(\mathbf{x})$ via Gaussian process regression[38]; (2) inferring the covariance matrix by applying a kernel averaging to residues $\mathbf{r}(\mathbf{x}_i) - \boldsymbol{\mu}(\mathbf{x}_i)$ (index $i$ represents the $i$-th data point). Kernel parameters are optimized to maximize data log-likelihood.

## GKR outperforms empirical estimation methods on synthetic datasets

We evaluated GKR on both a one-dimensional synthetic model (Fig. 2B–D) and a two-dimensional synthetic model (Supplementary Fig. 4). Each model consisted of a ground truth manifold, $\boldsymbol{\mu}(\mathbf{x})$, where each component represented a synthetic neural tuning curve, and a covariance matrix, $\Sigma(\mathbf{x})$. We generated data from these models using a Gaussian distribution (Eq. 1) and applied GKR to infer the manifold and covariance matrix.

For comparison, we also applied bin averaging and the Ledoit-Wolf (LW) method. In bin averaging, we discretized the label space $x$ into small bins, treating all data within a bin as having the same label. The sample mean and covariance within each bin served as estimates of the inferred manifold and covariance matrix. The LW method builds on bin averaging by incorporating shrinkage regularization to improve covariance estimation (see Methods)[43].

Using the inferred manifold and covariance matrix, we computed other geometric quantities, including the Riemannian metric, precision matrix, and Fisher information (see Methods). To assess the inference performance, we compared these inferred quantities to their ground truth values by computing the relative estimation error, defined as the difference between the estimate and the ground truth, normalized by the ground truth. Across various experimental conditions and in both one-dimensional and two-dimensional synthetic datasets, GKR consistently outperformed bin averaging and LW methods (Fig. 2B–D, Supplementary Fig. 4).

## Grid cell population activity manifold exhibits a toroidal-like topology

We then applied GKR to the grid cell sampled dataset $\mathscr{D}_s$. The inferred manifold is intrinsically three-dimensional, as it is parameterized by three label variables of interest: two spatial locations and one speed (notably, these three labels are largely uncorrelated, see Supplementary Fig. 5). To characterize the fitted manifold's topology, we conducted a persistent homology analysis (see Methods) and found that the manifold exhibits possibly toroidal topology (Supplementary Figs. 6A, B), consistent with previous findings[35].

Next, we examined how spatial locations were represented by grid cells. For each speed value, we defined a speed-slice manifold (SSM) as a cross-section of the full manifold, obtained by fixing speed while varying location (Fig. 3A). To visualize the SSM, we randomly sampled points from the manifold at a fixed speed of 20 cm/s, projected them onto the first six principal components (PCs) using principal component analysis (PCA), and further reduced the dimensionality to three using Uniform Manifold Approximation and Projection (UMAP)[44]. The resulting visualization (Fig. 3B), along with persistent homology analysis (Supplementary Fig. 6C), suggests that the SSM exhibits possibly toroidal topology.

## Running speed dilates the grid cell toroidal-like speed-slice manifold

A key question is how speed modulates the geometry of the SSM. Direct visualization of SSMs at different speeds is challenging, so we instead examined speed modulation using an example lattice on the SSM. This lattice consists of four spatially adjacent points, centered at the OF center, with an inter-point distance of 2 cm (Fig. 3C). While keeping these four spatial locations fixed, we varied the speed components to construct a lattice manifold. PCA analysis revealed that this manifold is low-dimensional, with three principal components accounting for over 90% of the variance (Supplementary Fig. 7). Based on this, we projected the lattice manifold into three or two dimensions for visualization (Fig. 3D). The results indicate that the lattice expands as speed increases.

In addition to the lattice manifold, we also visualized other manifold slices, including those obtained by fixing the rat's x-coordinate and

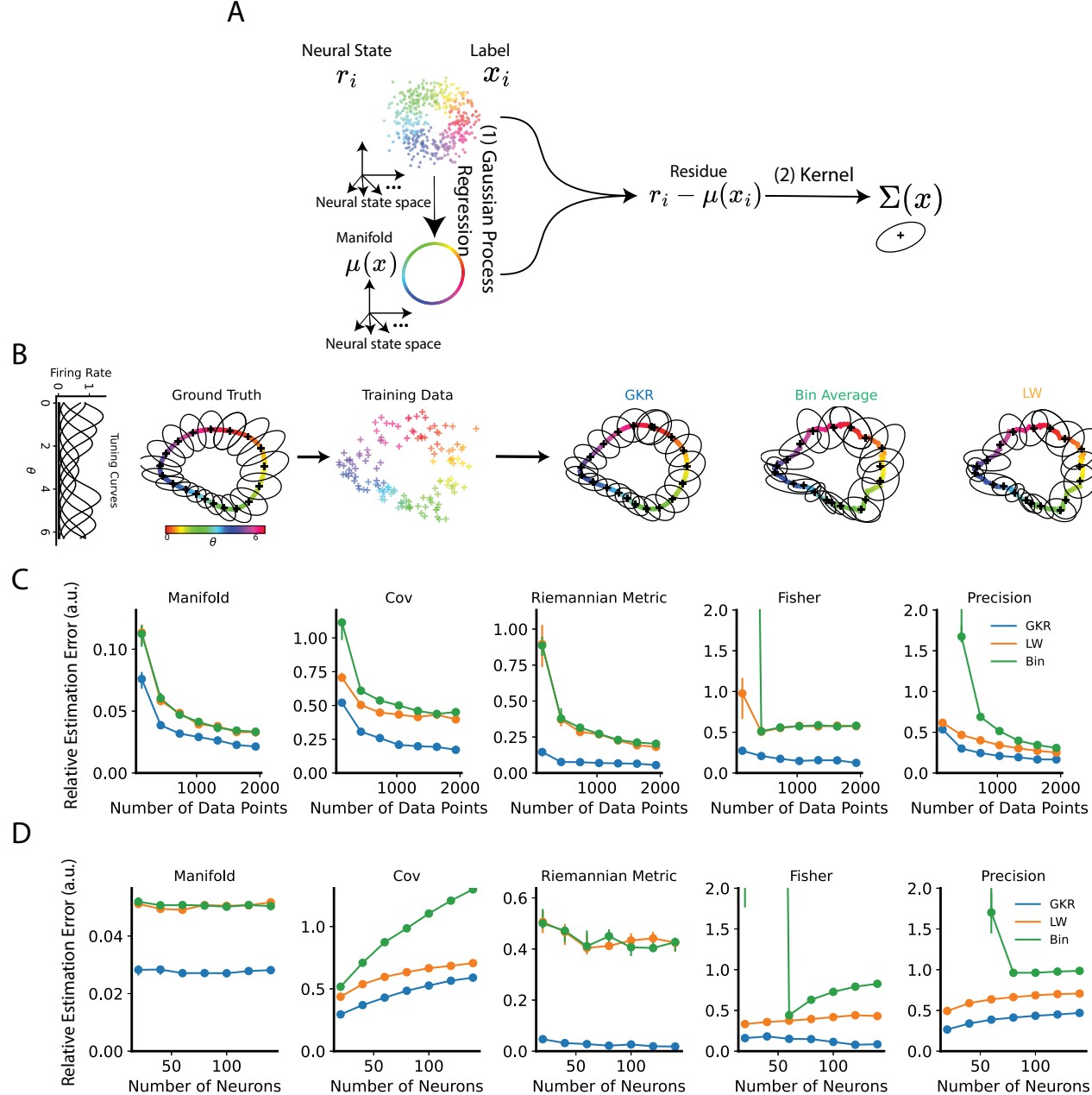

**Fig. 2 | Gaussian Process with Kernel Regression (GKR) infers the smooth manifold and covariance from noisy data. A** Given neural states **r** and labels **x**, the goal of GKR is to infer the conditional distribution $p(\mathbf{r}|\mathbf{x})$ as a smooth function of **x**. GKR approximates $p(\mathbf{r}|\mathbf{x})$ as a Gaussian distribution and separates the inference problem into two steps: inferring mean $\boldsymbol{\mu}(\mathbf{x})$, and inferring noise covariance $\Sigma(\mathbf{x})$. **B** Application of GKR to a synthetic dataset. The synthetic dataset consists of $N$ synthetic neurons with heterogeneous tuning curves to a circular label $\theta$ ranging from 0 to $2\pi$. The ground truth $\boldsymbol{\mu}(\theta)$ and $\Sigma(\theta)$ are visualized in the first two principal components plane (via PCA, left panel). Ellipses indicate the principal axes of covariance, with lengths proportional to the eigenvalues. In this example, the dataset consists of 10 neurons and 100 data points, which were used to fit manifolds via GKR, bin averaging, and Ledoit-Wolf (LW) methods, respectively (right panel; see Methods). The default dataset consists of 300 data points and 10 neurons, with the number of data points (**C**) or neurons varying (**D**) as indicated by x axes. Dataset was used for estimating geometric metrics. The estimated geometric metrics were compared against ground truth values and evaluated using relative estimation error, defined as the normalized difference between the estimated and true values. Dots and error bars represent the median, first, and third quartiles of relative estimation error across 10 randomly generated synthetic datasets (see Methods). A similar analysis for a 2D synthetic manifold is presented in Supplementary Fig. 4. Source data are provided as a Source Data file.

using a larger lattice. All visualizations consistently imply that the SSM dilates with increasing speed (Supplementary Fig. 7).

To quantify changes in SSM size, we used two metrics: (1) SSM radius, defined as the average distance from the SSM surface to its center, providing a global measure of SSM size; and (2) Lattice area, computed as the area of a parallelogram whose sides are tangent vectors of the SSM, capturing local manifold surface size. Across all experimental configurations examined, both SSM radius and lattice area increase with running speed, indicating that the grid cell SSM dilates as speed increases (Fig. 3E).

## Running speed increases grid cell population noise
SSM dilation intuitively enhances spatial coding. Consider a binary classification task distinguishing two neural state classes

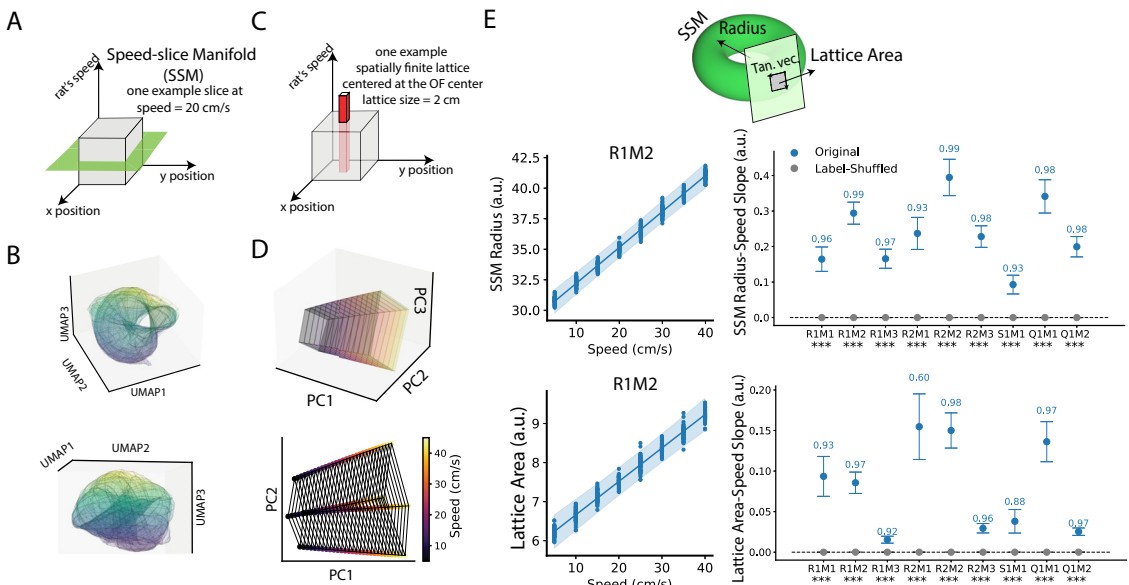

**Fig. 3 | Speed dilates the grid cell population toroidal-like manifold. A** A speed-slice manifold is a cross-section of the full manifold, obtained by holding speed constant while varying location. **B** We visualized an example SSM (speed = 20 cm/s) by first projecting it onto the first six PCs, then further reducing it to three latent dimensions using UMAP[44]. Color represents the third UMAP component value only for better visualization. The upper and bottom panels show two views of the same SSM. **C** We also visualized the representation of four fixed spatial positions (i.e., lattice) but varying speed values. **D** For visualization, the lattice manifold was projected into the first three PCs (upper) and two PCs (bottom) respectively (see Supplementary Fig. 7 for cumulative variance explained ratio). **E** SSM size was measured in the original high-dimensional space (dimension equals the number of neurons). SSM radius is the average distance from points on the manifold to the

manifold center. The lattice area measures the parallelogram area formed by two tangent vectors (i.e., Tan. vec., differentiated along x and y labels respectively, see Methods). Left: Each dot represents the measured quantity from one sampled dataset $\mathscr{D}_s$ at one speed bin (see Methods, fifty sampled $\mathscr{D}_s$ with eight speed bins = four hundred data points). The solid line and error band show the best-fitting line and 95% CI estimated using BLEA. Right: Dots and error bars represent mean and 95% CI of the slopes estimated using BLEA (left panel shows the case of R1M2). The numbers above error bars are r-squared scores. The texts below x axis tick label represent the significance level whether the slope fitted from original data $\mathscr{D}_s$ differs from that fitted from label-shuffled data (two-sided, Bayesian method, see Methods); ***$p < 0.001$; **$p < 0.01$; *$p < 0.05$; NS not significant. Source data are provided as a Source Data file.

---

corresponding to nearby locations (e.g., Fig. 1B). By increasing the separation between these classes, SSM dilation makes their representations more distinguishable. However, discriminability is not solely determined by distance—noise strength also plays a crucial role. Increased noise in the grid cell population reduces spatial coding accuracy. This raises a question: what's the effect of running speed on grid cell population noise?

To investigate this, we fitted manifolds and noise covariances for the sampled datasets $\mathscr{D}_s$ using GKR. Total noise is the trace of the covariance matrix (Fig. 4A). We found that total noise increases with increasing speed (Fig. 4B, C). Compared to total noise, noise projected onto the manifold may be more relevant to information coding[26]. Therefore, we projected the covariance matrix onto the tangent plane of the SSM, and computed the trace of the projected covariance matrix as the projected noise. Consistent with total noise, projected noise also increases with speed (Fig. 4B, C).

### Fisher information increases with increasing speed, indicating that the effect of speed-slice manifold dilation outpaces the effect of increasing noise

Running speed has opposing effects on spatial coding. On the one hand, it expands the smooth spatial manifold (SSM), pushing neural representations of nearby locations further apart (Fig. 3E), thereby improving spatial coding. On the other hand, it increases grid cell population noise (Fig. 4C), which degrades spatial coding. To assess the overall impact, we examined (linear) Fisher information, a metric that quantifies the local discriminability of neural population representations, defined as $(\partial\boldsymbol{\mu}/\partial\mathbf{x})^T\Sigma^{-1}(\partial\boldsymbol{\mu}/\partial\mathbf{x})$, which incorporates both the noise factor ($\Sigma$) and the lattice area factor (lattice area is formed by tangent vectors $\partial\boldsymbol{\mu}/\partial\mathbf{x}$) (Fig. 5A). Fisher information is a commonly

used metric for assessing the local discriminability of neural population representations. The total Fisher information, given by the trace of the Fisher information matrix, measures the overall precision of the representation, with higher values indicating better spatial coding[45].

It is well known that Fisher information is hard to estimate in a high-dimensional space[24]. Therefore, besides using the original $\mathscr{D}_s$, we also projected $\mathscr{D}_s$ into the first six PCs, denoted as $\mathscr{D}_s^{(6)}$. Each $\mathscr{D}_s^{(6)}$ was then fed into GKR for fitting a GKR model. GKRs fitted from both $\mathscr{D}_s$ and $\mathscr{D}_s^{(6)}$ were analyzed identically to double-check our results on Fisher information.

We computed the total Fisher information from the fitted GKRs (see Methods, Supplementary Fig. 8A for $\mathscr{D}_s$ and Fig. 5A for $\mathscr{D}_s^{(6)}$). Slope analyses indicate that Fisher information increases with running speed in both $\mathscr{D}_s$ and $\mathscr{D}_s^{(6)}$ (Fig. 5B for $\mathscr{D}_s^{(6)}$, Supplementary Fig. 8B for $\mathscr{D}_s$, although four out of nine experimental configurations in the results of $\mathscr{D}_s$ do not show statistical significance, possibly due to the curse of dimensionality). This increase in Fisher information with running speed suggests that the effect of SSM dilation outweighs that of increasing noise, leading to improved spatial coding at higher speeds, which is qualitatively consistent with the results obtained using SCA (Fig. 1B–D).

Fisher information, derived from a purely bottom-up geometric approach, in fact is intrinsically linked to SCA (Fig. 1B). Specifically, we established a theoretical upper bound for SCA directly from Fisher information (see Methods). This bound is approximately a linear function of the square root of total Fisher information. Therefore, an increase in Fisher information implies a corresponding increase in SCA.

We first tested this theoretical upper bound in synthetic datasets (Supplementary Fig. 9). We showed that the SCA computed directly

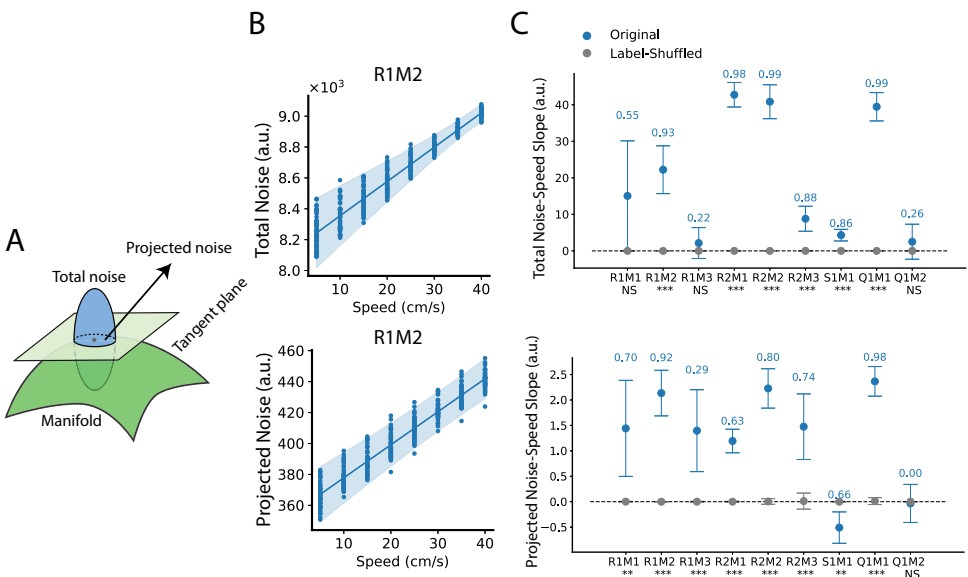

**Fig. 4 | Running speed increases grid cell population noise. A** Total noise is the trace of the covariance matrix. Projected noise is the trace of the covariance matrix projected onto the SSM tangent plane. **B** Each dot is the measured quantity from one sampled dataset $\mathscr{D}_s$ at a speed value (see Methods, fifty sampled $\mathscr{D}_s$ with eight speed bins = four hundred data points). The line and error band show the best linear fitting line and 95% CI using BLEA. **C** Dots and error bars represent mean and 95% CI of the slopes estimated using BLEA (panel B shows the case of R1M2). The numbers above error bars are $r$-squared scores. The texts below $x$ axis tick label represent the significance level whether the slope fitted from original data $\mathscr{D}_s$ differs from that fitted from label-shuffled data (two-sided, Bayesian method, see Methods); ***$p < 0.001$; **$p < 0.01$; *$p < 0.05$; NS not significant. Source data are provided as a Source Data file.

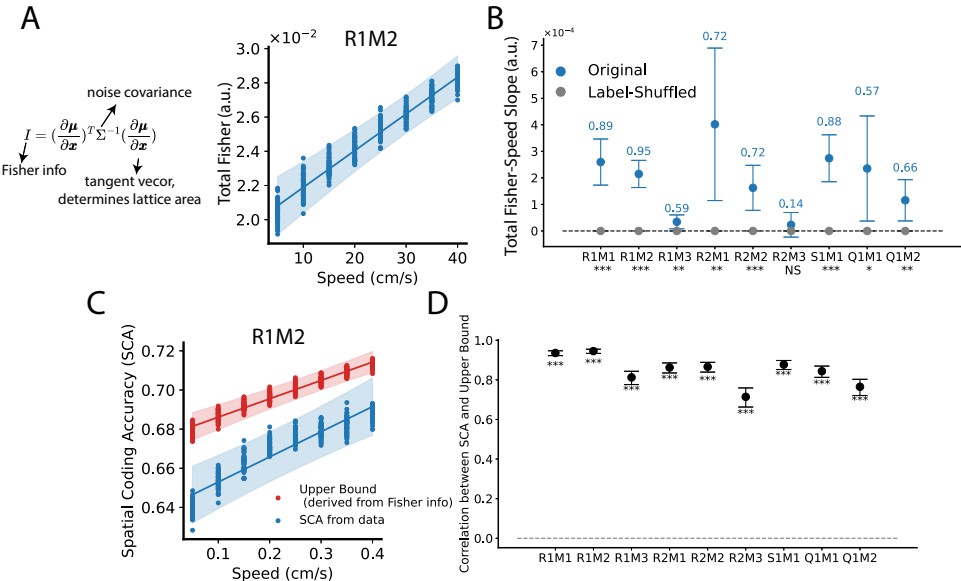

**Fig. 5 | Grid cells' (linear) Fisher information increases with increasing running speed. A** (Linear) Fisher information mathematically combines noise covariance and tangent vectors (which determine lattice area), which is commonly used to measure the local discriminability of information from noisy neural states[45]. Total Fisher information is the trace of the Fisher information matrix. Each dot represents the measured total Fisher information from one dimensionally reduced sampled dataset $\mathscr{D}_s^{(6)}$ ($\mathscr{D}_s$ projected into its first six PC subspace) at a specific speed value (fifty sampled $\mathscr{D}_s^{(6)}$ with eight speed bins = four hundred data points). The line and error band show the best linear fit and 95% CI using BLEA. **B** Dots and error bars show the mean and 95% CI of the estimated slopes using BLEA (see Methods, panel A shows the case of R1M2). The numbers above the error bars represent r-squared scores. The texts below $x$-axis tick labels indicate the significance level of whether the slope fitted from the sampled dataset $\mathscr{D}_s^{(6)}$ differs from that of label-shuffled data (two-sided, Bayesian method, see Methods); ***$p < 0.001$; **$p < 0.01$; *$p < 0.05$;

NS not significant. **C** We computed theoretical SCA upper bounds derived from the Fisher information (red, see Methods), and also showed the actual SCA computed directly from data (blue, similar as Fig. 1B, C, but using $\mathscr{D}_s^{(6)}$ rather than $\mathscr{D}_s$). Dots and error bands have the same meaning as in (**A**). **D** Correlation between upper bounds and SCA. For each experimental configuration, fifty $\mathscr{D}_s^{(6)}$ and eight speed bins were used, thus four hundred data points for upper bound and actual SCA, respectively (see **C**). We then computed the correlation (black dots) between these two sets of four hundred data points, with error bars indicating the 95% CI obtained via Fisher transformation[87]. The texts below the error bars indicate the significance levels of whether the correlation differs from zero: ***$p < 0.001$; **$p < 0.01$; *$p < 0.05$; NS not significant (Pearson correlation test, two-sided). The results of the same analysis but using the original datasets $\mathscr{D}_s$ are similar (Supplementary Fig. 8). Source data are provided as a Source Data file.

from the synthetic datasets is well bounded by the theoretical upper bound predicted by the Fisher information. Moreover, the upper bound exhibits trends consistent with actual SCA across different dataset parameters (e.g., number of data points, dimensionality).

We then applied this analysis to grid cell datasets, computing the theoretical SCA upper bounds using Fisher information fitted from GKR. The computed upper bounds are well above the actual SCA values (Fig. 5C, Supplementary Fig. 8C). More importantly, both SCA and its upper bound exhibit similar speed modulation effects. The correlations between the predicted upper bounds and actual SCA are positive (Fig. 5D, Supplementary Fig. 8D).

Overall, Fisher information, derived from the geometric properties of the SSM and noise, quantitatively aligns with decoding performance measured via SCA. Both approaches support the conclusion that grid cell spatial coding improves with increasing running speed.

### Results from GKR agree with those from a modified method that does not assume data normality

GKR approximates the distribution $p(\mathbf{r}|\mathbf{x})$ as a normal distribution, which might not always valid (see Methods and Supplementary Fig. 10). To verify the validity of our previous sections' results, we propose a modified approach—Gaussian process regression with kernel sampling (GKR-S)—that does not assume data normality. GKR-S contains two steps: first, it uses Gaussian process regression to estimate the manifold (as in GKR); second, it resamples from neighboring labels $\mathbf{x}_i$ to generate pseudo-samples and thereby estimate the conditional distribution (see Methods).

We evaluated GKR-S on a synthetic dataset and found that it performs well only for low-dimensional data (Supplementary Fig. 11A). Therefore, we projected the grid cell datasets onto their first six PCs and applied both GKR-S and GKR to estimate geometric properties (e.g., manifold size, Fisher information etc.). The two methods yielded closely matching results, both revealing better spatial representation at higher running speeds, indicating that, despite its normality assumption, GKR remains a reasonable method to estimate these geometric properties (Supplementary Fig. 11).

### Speed modulation effects on grid cell information geometry can be qualitatively reproduced by a simulated independent Poisson Speed-Gain (IPSG) grid cell population

What do these population results imply about individual grid cells? Here, we propose a simple model—the Independent Poisson Speed-Gain (IPSG) grid cell population, which contains three key assumptions: (1) grid cell firings are independent; (2) grid cells are Poisson neurons; and (3) the effect of running speed on individual grid cell firing is a monotonically increasing gain factor[11].

Using these assumptions, we derived analytical expressions for manifold size, noise strength, and Fisher information as functions of running speed. Both these analytical results and additional numerical simulations demonstrate that the IPSG can qualitatively reproduce the observed positive speed modulation effects in Figs. 3E, 4C, 5B (Supplementary Fig. 12, see Methods).

Intuitively, in IPSG, increasing speed amplifies each cell's firing without altering its spatial tuning. Because manifold size scales roughly with firing rate, it too grows with speed (see Methods, Fig. 3E, Supplementary Fig. 12). Poisson-neuron assumption implies that noise (the standard deviation) scales as the square root of the firing rate, so noise increases more slowly than firing rate. As a result, each cell's signal-to-noise ratio—and hence its Fisher information—rises with speed under the independent-firing assumption (see Methods). More loosely, IPSG suggests that faster running increases grid-cell firing (without disrupting the rate map too much) more than noise and the effects of noise correlations are negligible, explaining the observed speed modulations on the manifold (Figs. 3E, 4C, 5B). Whether this picture holds in real grid-cell data remains unclear. We leave more precise

theoretically modeling and validations for future work, while this paper focuses more on data-driven descriptive analysis.

### Grid cell activity noise correlation is information-detrimental

Our results indicate that grid cell spatial coding improves at high speeds, based on our analysis of simultaneously recorded grid cell population activity. A key advantage of analyzing population activity, as opposed to individual neural responses, is that it inherently accounts for the effects of noise correlation on spatial coding. Here, we use the term 'noise correlation' specifically to refer to the cell-to-cell noise covariance (i.e., between two different cells). Noise correlation can be information-beneficial or information-detrimental, depending on the geometric relation between noise covariance and the information encoding manifold (Fig. 6A, also see a two-neuron toy example as an illustration in Supplementary Fig. 13)[24,26]. In this section, we explicitly examine how noise correlation influences grid cell spatial coding.

To explore the role of noise correlation in spatial coding, we compared outcomes from the original grid cell dataset ($\mathscr{D}_s$) with those from a hypothetical population of independent firing grid cells (IFGC)[26]. A conventional method to eliminate noise correlation is trial shuffling. For neural states recorded under identical conditions (e.g., the same spatial location) across different trials, one can randomly permute the firing profile of each neuron across trials. This shuffling preserves single-cell statistics while effectively disrupting intercellular noise correlation.

Although the OF task does not include repeated trials, the GKR serves as a generative model that can produce multiple data points for each condition. Specifically, for a given condition $\mathbf{x}$, the fitted GKR generates (theoretically) an infinite number of data points drawn from a Gaussian distribution (Eq. 1). After the trial-shuffling procedure is applied to these generated data points, the mean remains unchanged, but the covariance matrix becomes diagonal, with all off-diagonal elements set to zero. Thus, the IFGC's GKR model is equivalent to the original GKR model with a purely diagonal covariance matrix.

We computed the total noise levels for both the GKR and IFGC GKR models (Fig. 6B). As expected, removing the off-diagonal elements of the covariance matrix does not alter its trace, leaving the overall noise unchanged. However, this manipulation affects the projected noise: the GKR model exhibits a higher projected noise than the IFGC GKR model (Fig. 6C). This finding indicates that cell-to-cell noise correlations in grid cell activity "reshape" the noise structure, directing a larger fraction of noise onto the torus surface. Consistent with this, increasing noise correlation leads to smaller Fisher information (Fig. 6D), underscoring their detrimental impact on information encoding.

To further validate that the correlation is information-detrimental, we employed a top-down decoding approach based on linear classification accuracy of neural states from two spatial boxes (i.e., SCA in Fig. 1B), without using GKR. For the IFGC SCA, we randomly permuted each neuron's firing rates within each box. This procedure preserves the single-cell statistics while effectively disrupting the noise correlations. Analysis of the permuted data showed that the IFGC SCA exceeded the SCA from the original datasets (Fig. 6E), thereby confirming that noise correlation is information detrimental.

## Discussion

Accurate internal spatial representation is essential for navigation, and grid cells are widely regarded as a fundamental component of this process[2,3]. Previous analyses of speed modulation on grid cell coding have predominantly focused on individual cells or cell pairs[11–14], thereby neglecting the influence of population noise covariance—a factor that can significantly impact coding fidelity[26]. Here, we developed GKR to study the population coding from an information geometry perspective. We demonstrated that the grid cell manifold

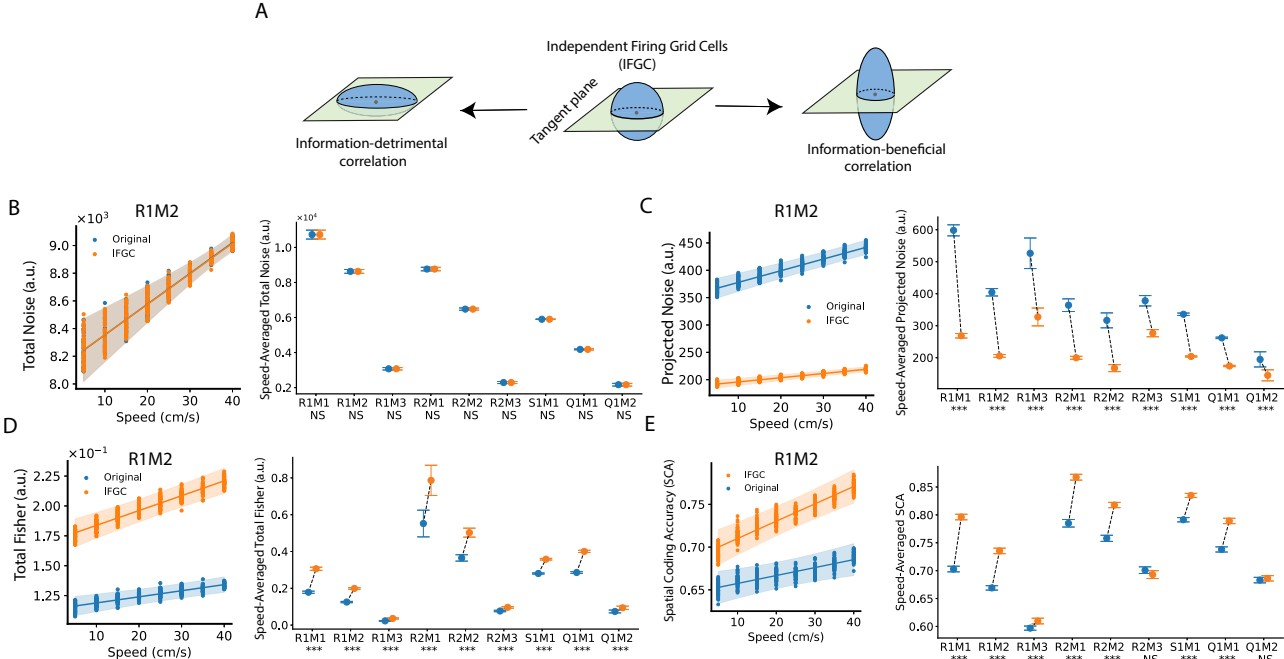

**Fig. 6 | Noise correlation in grid cells increases noise projection onto the manifold and impairs information coding. A** Noise correlation can either enhance ("information-beneficial") or impair ("information-detrimental") coding relative to a model of independent firing grid cells (IFGC). **B** IFGC's GKR is identical to the GKR fitted from the original sampled dataset $\mathcal{D}_s$, except that the off-diagonal elements of the covariance matrix are set to zero (see texts and Methods). Total noise is the trace of the covariance matrix. Left: Each dot represents the total noise from one GKR model fitted to a specific $\mathcal{D}_s$ at a given speed (fifty sampled $\mathcal{D}_s$ with eight speed bins = four hundred data points). The lines and error bands show the best linear fitting and 95% CI using BLEA. Right: We computed the speed-averaged total noise as the average total noise across all speeds (from 5 cm/s to 45 cm/s) per $\mathcal{D}_s$. Since fifty $\mathcal{D}_s$ were used, we have fifty speed-averaged total noise values per experimental configuration, allowing fitting a normal distribution. Dots and error bars show the mean and 95% CI of the estimated speed-averaged total noise distributions (see Methods). The texts below each x-axis tick label indicate the

significance level of whether the speed-averaged total noise fitted from the original GKR differs from that of IFGC GKR (two-sided, Bayesian method, see Methods). ***$p < 0.001$; **$p < 0.01$; *$p < 0.05$; NS not significant. **C**, **D** Same as (B), but measuring projected noise and total Fisher. Statistical testing (Bayesian Methods) on projected noise (or total Fisher) is one-sided—whether the projected noise (or total Fisher) obtained from the original GKR is greater (or smaller) than IFGC GKR (see Methods). **E** The key idea of SCA is to assess classification accuracy between data points drawn from two spatial boxes. To compute IFGC's SCA, we generated a "trial-shuffled" dataset by permuting each cell's firing activity across all data points within the same box, allowing eliminates intercellular noise correlations. Illustrations are the same as panel B, except showing SCA at different speeds. Statistical testing is one-sided (Bayesian method, see Methods), indicating whether the SCA from the original dataset is smaller than IFGC's SCA. Source data are provided as a Source Data file.

expands in size as speed increases. This manifold dilation effect exceeds the increase in noise, as indicated by the higher Fisher information observed at high speeds. Overall, our results favor the hypothesis that increasing running speed increases grid cell spatial coding accuracy. GKR can be a powerful tool to study neural population representation from an intuitive information geometric perspective.

However, GKR has its limitations. First, it does not perform well in very high-dimensional spaces, which may require a larger number of data points (Fig. 2 and Supplementary Fig. 4). This issue may be mitigated by first applying a dimensionality reduction method to the data[46]. Second, GKR assumes that the data follows a normal distribution. While the normal distribution is a commonly used approximation[34,38]—GKR may produce unreliable results if the true distribution deviates significantly from normality. It is advisable to perform a normality test (Supplementary Fig. 10), compute test data's likelihood[34], or use an alternative method (e.g., GKR-S, Supplementary Fig. 11) to verify the results. Finally, GKR is only applicable when the data explicitly contains labels. In other words, its purpose is to evaluate the geometric representation properties of known labels of interest. For example, in vision[42], navigation, or working memory[47] studies, the labels of interest are often defined by stimulus parameters. However, in cases where one aims to evaluate the representation structure of latent variables, it is necessary to first apply latent variable inference methods[46,48], and then apply GKR. Overall, future improvements to

GKR could focus on enhancing its performance in high-dimensional settings, adapting it for non-normal data, and extending its applicability to scenarios without explicit labels.

We analyzed the population-level properties of grid cell activities, but what are the implications of these findings for individual grid cells? There are two major models of grid cells[49]: the rate-based model and the oscillatory-interference model. First, in some rate-based models (continuous attractor networks[20]), running speed serves as an input to the grid-cell network. Faster speeds may elevate firing rates[11,12], which can sharpen the spatial rate map, enhance the signal-to-noise ratio, and thus boost population Fisher information (Fig. 5). Although some other rate-based models contain normalization mechanisms implying the opposite—that the population mean firing rate should not change with speed (e.g., self-organizing models[50]). Secondly, in the oscillatory-interference model, running speed modulates the frequency of membrane potential oscillation which might lead to more accurate spatial fields[49] (although a study suggests that MEC theta frequency is modulated by acceleration rather than speed[51]). Finally, higher running speeds imply more frequent encounters with environmental boundaries, allowing for more frequent corrections in grid coding[18]. Despite these conjectures, it should be noted that real grid cells are complicated, involving correlated noise. Models based solely on individual grid cells, without accounting for noise correlations, may result in substantial estimation errors, as shown by the pronounced discrepancies between the original data and the IFGC model in Fig. 6. The

connection between population results and individual grid cells remains for future exploration.

Rats typically exhibit higher running speeds in novel environments[52], as implied by this paper, which might enhance grid coding thus supporting more effective adaptation to novel surroundings. Grid cell representations are known to change in novel environments[53–55]. Some studies suggest that the grid pattern rescales in a novel environment (e.g., Barry et al. 2012[56]); others propose that rats refine their grid coding by learning the environment's boundaries[18,55]. These alterations in individual grid cell patterns may reflect corresponding changes in the representation geometry, such as a rescaling of the toroidal structure or localized distortions near environmental boundaries. The effects of environmental modulation on population-level representations remain an open question for future investigation.

Beyond grid cells, how does running speed influence other cell types in the navigation system? Hardcastle et al. found that the spatial decoding accuracy of MEC neurons improves at higher speeds, suggesting that increased speed generally benefits MEC spatial representation[10]. This aligns with findings that, similar to grid cells, running speed predominantly increases the firing rates of other MEC cell types, including head direction cells, speed cells, and conjunctive cells[11,17]. However, the modulation effects of running speed on hippocampal cells may be more complex. Grid cells are modeled as a primary feedforward input to the hippocampus[8,57], suggesting that running speed should also enhance place cell representation as well. However, this feedforward model is a simplification, as the hippocampus sends feedback projections to the MEC[58]. Moreover, place cells receive inputs not only from grid cells but also from other sources, such as head direction cells[57]. These additional mechanistic factors obscure how running speed modulates place cell activity. Indeed, earlier research suggests that the majority of place cells are not strongly modulated by running speed, at least not in an obvious manner[59]. Nevertheless, beyond speed modulation, movement direction could influence the representational geometry of place cells, as it has been shown to reshape place fields[60]. Extending geometric analyses to other cell types remains an interesting avenue.

Running speed modulation effects have been widely observed across other brain regions as well[61]. For example, locomotion primarily suppresses neural activities in the auditory cortex[62,63]. In contrast, locomotion generally enhances V1 neuron activity[64,65], but may turn to suppression after certain high running speeds[66]. In fact, the effect of locomotion modulation is usually entangled with other modulation factors[61,66,67]. For example, V1 neural activities are jointly influenced by both animal's running speed and visual stimuli movement speed[65]. This influence can be mathematically expressed as a weighted sum of the two speed contributions, with weights varying diversely across neurons. The geometric approach has been shown to be a practically effective method to assist in understanding the diversity of individual neurons from a comprehensive population-level perspective[41,68,69]. GKR can be a useful tool to understand the diversity of running speed modulations in different brain areas.

One advantage of GKR is its ability to provide detailed inspection of location geometry. Local geometry reveals the intricacies of information coding within a small range of values, which is particularly useful for comparing the representation bias of different information values. Representation bias has been observed in the navigation system[60,70]. For instance, place and grid cells' fields tend to shift towards reward locations, which has been interpreted as an over-representation of rewarded locations[70–73]. From a geometric perspective, overrepresentation implies larger Fisher information, which can be attributed to either local manifold dilation, reduced projected noise, or both (Figs. 3, 4, 5). The concepts of local manifold dilation and reduced noise have been supported in working memory studies: (1) The working memory system may use attractors to reduce noise[47,74,75].

(2) Recurrent neural networks (RNNs) trained on working memory tasks utilize larger state spaces to represent common values, thus yield improved Fisher information[47]. In RNNs, manifolds are often observed to be quite simple, usually taking the form of a low-dimensional ring structure[47]. This simplicity allows the size of the encoding space to be measured using straightforward methods. However, in the actual brain, manifolds can be highly complex and high-dimensional[69]. The GKR method illustrated in this paper can be particularly helpful in studying the local structure of these complex, high-dimensional manifolds, assisting the analysis of representation bias.

## Methods

### Experimental data
Experimental data were collected by Gardner et al.[35]. Rats performed open-field foraging (OF) tasks in a 150 cm wide OF box. Three-dimensional motion capture tracked the rats' head positions and orientations using five retroreflective markers attached to the implant during recordings. The 3D marker positions were then projected onto the horizontal plane to determine the rats' 2D positions. Neuropixel probes recorded neural activity in the MEC. Neural activity were then processed using a clustering method to classify neurons into grid cells and non-grid cells[35]. In total, these procedures yielded nine sets of simultaneously recorded grid cell population activities (i.e., nine experimental configurations): rat 'R' day 1 modules 1, 2, 3; rat 'R' day 2 modules 1, 2, 3; rat 'S' module 1; and rat 'Q' modules 1, 2. We used a shorthand notation, e.g., "R1M2", to represent rat R ("R") on day 1 ("1") and grid cell module two ("M2"). Note that "R1" does not necessarily refer to the same day as "S1". Day labels are used solely to distinguish recordings from the same rat. These processed data are available from Gardner et al. 2022[35].

### Grid cell rate map
The grid cell rate maps shown in Fig. 1A and Supplementary Fig. 1 were computed as follows. Firing rate was estimated by dividing spike counts by 10-ms time bins and then convolving the result with a Gaussian filter with a standard deviation of 20 ms. To estimate the averaged firing rate at different locations, the OF box (150 × 150 cm) was digitized into small spatial bins of 3 × 3 cm. Firing rates at each visited spatial bin were averaged, and those at each unvisited bin were set to 0. To correct the effect of unvisited bins, we created a mask ($M_0$) with a value of 1 at the visited bins and 0 at unvisited bins. Next, both the firing rate and mask $M_0$ were spatially convolved with a 2D Gaussian filter with a standard deviation σ = 8.25 cm. The convolved firing rate was divided by the convolved $M_0$ to obtain the final corrected rate map for each cell.

### Gridness
Gridness measures how well a grid cell's rate map conforms to a hexagonal pattern[12]. Some grid cells' rate maps have incomplete peaks at the OF box boundaries. To correct this boundary effect, the rate map was first padded by 30 cm on each side. This padding was performed by linearly ramping the firing rate at the edges to zero over the outer 30 cm of the padded area. (implemented using the 'numpy.pad' function in Python, with 'mode = 'linear_ramp''). Autocorrelating the padded rate map produced an autocorrelation map. The boundary effect of the autocorrelation was corrected by padding with zeros on all sides (implemented using 'scipy.signal.correlate2d (padded_rate_map, padded_rate_map, mode = 'same', boundary = 'fill', fillvalue = 0)').

The autocorrelation map was masked by two circles centered at the map's center. The outer circle's diameter matched the edge length of the autocorrelation map. The inner circle's area was 15% of the outer circle's area to filter out center peaks on the map. Only the regions between the two circles were kept; the rest were set to 0. Next, the masked autocorrelation map was correlated with its rotated versions (rotated by 30, 60, 90, 120, and 150 degrees, respectively). A well-defined grid cell should have peak correlation values at 60 and 120 degrees, and valleys at 30, 90, and 150 degrees. Gridness was

calculated by subtracting the average valley values (30, 90, and 150 degrees) from the average peak values (60 and 120 degrees).

## Data preprocessing

Time was binned in 10-ms intervals. Spikes count at each time bin was computed, and then divided by 10 ms as an estimate of firing rate. The firing rate was then temporally smoothed using a Gaussian kernel with a standard deviation of 20 ms. To estimate the rat's speed, velocity was first computed by computing the finite differences of the rat's positions, i.e., $(\mathbf{p}_{i+1} - \mathbf{p}_{i-1})/20$ where $\mathbf{p}_i$ is the rat's position at time bin $i$. The velocity's L2 norm is the speed. This procedure provided a feature map indicating grid cell firing rates, with rows representing time bins and columns representing grid cell IDs; and a label with rows representing time bins and three columns indicating $x$ location, $y$ location, and speed. Data with speeds lower than 5 cm/s or higher than 45 cm/s were excluded. Grid cells with low gridness (below 0.1) were also excluded. The combined feature map and label are termed as a grid cell dataset, denoted as $\mathscr{D}$. There are 9 grid cell datasets corresponding to different experimental configurations (different rats, grid cell modules, and different days). The number of grid cells in each dataset is: 113 in R1M1, 132 in R1M2, 51 in R1M3, 140 in R2M1, 153 in R2M2, 62 in R2M3, 96 in S1M1, 81 in Q1M1, 53 in Q1M2.

The speed distribution in $\mathscr{D}$ is highly biased. It has more data in low-speed region than in the high-speed region (Supplementary Fig. 2). This biased distribution of data may cause potentially biased evaluation. To avoid this, we performed resampling on the dataset as follows. Speeds ranging from 5 cm/s to 45 cm/s were binned into 5 cm/s bins. Data in each speed bin was collected. Let $N_{sp}^{\min}$ denote the minimum number of data points among all speed bins, we defined $K$ as the $\min\{N_{sp}^{\min}, 10,000\}$. In each speed bin, we sampled $K$ data points (without replacement). Sampled data points from different speed bins were combined to create a single sampled dataset, denoted as $\mathscr{D}_s$. $\mathscr{D}_s$ has a roughly equal amount of data at each speed value. The above sampling procedure was repeated 50 times, resulting in 50 sampled datasets $\mathscr{D}_s$ per experimental configuration.

As a baseline comparison, we also shuffled the data $\mathscr{D}$ by permuting the label timestamps, thereby disrupting the relationship between neural states and labels. This permuted data was then processed using the same sampling procedure as described above, yielding 50 label-shuffled-sampled datasets.

The dimensionality of $\mathscr{D}_s$ is the number of grid cell, which can be more than 100. This can pose a challenge in accurately estimating covariance and Fisher information[24]. Therefore, we also performed PCA on $\mathscr{D}_s$, projecting onto the first 6 principal components to obtain $\mathscr{D}_s^{(6)}$. The same projection procedure was applied to the shuffled datasets. These projected datasets were used for estimating Fisher information in Fig. 5, and comparing to GKR-S in Figure 11.

## Spatial coding accuracy

A common way to evaluate the quality of neural population representation is to assess how accurately a simple linear classifier can distinguish between neural population representations of two adjacent experimental conditions (e.g., stimulus parameters or locations in this paper)[22]. In this paper, this type of classification accuracy is referred to as spatial coding accuracy (SCA, Fig. 1B).

For a given sampled dataset $\mathscr{D}_s$, we split it into 8 speed-split datasets (SSD) based on speed values. Specifically, data with speed values within $[v_i, v_i + 5\text{cm/s}]$ were collected as one SSD, where $v_i = 5, 10, \ldots, 40\text{cm/s}$. For each SSD, we randomly sampled 300 spatial locations $\mathbf{x}_c$. For each location $\mathbf{x}_c$, we constructed two adjacent locations $\mathbf{x}_{\pm} = \mathbf{x}_c \pm \delta l \hat{\mathbf{e}}$, where $\hat{\mathbf{e}}$ is a unit vector with a random angle, $\delta l = 5\text{cm}$. Each $\mathbf{x}_{\pm}$ defines a small spatial box, centered at $\mathbf{x}_{\pm}$ with an edge length of 10 cm. Data from the two boxes were collected. To ensure fair classification, the data from the box with the larger number of data points were subsampled (without replacement) so that both

boxes had an equal number of data points. The data from the two boxes were then concatenated. If the total number of data points was less than 50, this $\mathbf{x}_c$ was discarded due to insufficient data. Otherwise, the concatenated data was split into train and test sets (0.67:0.33). A logistic classifier (with an L2 regularization coefficient C = 1, using the scikit-learn package) was then trained on the train set and evaluated on the test set. The classification accuracy averaged across all valid $\mathbf{x}_c$ is defined as the SCA of that speed bin $[v_i, v_i + 5\text{cm/s}]$. This procedure was applied to all speed bins, $\mathscr{D}_s$ (or $\mathscr{D}_s^{(6)}$ in Fig. 5C), and the label-shuffled dataset.

Besides using logistic regression for classification (Fig. 1), we also computed SCA using perception and support vector machines, see the results in Supplementary Fig. 3.

## Bayesian linear ensemble averaging and statistical testing

Metrics considered in this paper include SCA (e.g., Figs. 1C, D, 5C), torus radius, lattice area (e.g., Fig. 3E), total and projected noise (e.g., Fig. 4B, C), and Fisher information (e.g., Fig. 5, Supplementary Figs. 8A, B), etc. For each sampled dataset $\mathscr{D}_s$, we computed the metric values at different speed bins, forming a metric-speed dataset consisting of metric value $t_i$ and the corresponding speed value $v_i$, where $i$ indexes the $i$ th data point in the metric-speed dataset. For example, one dot in Fig. 1C is one data point in the SCA-speed dataset (with a corresponding $\mathscr{D}_s$). For convenience, we define $\mathbf{x}_i = (v_i, 1)$, which includes speed and a constant for a bias parameter. Currently, we limit our discussion to one $\mathscr{D}_s$, and later we will ensemble results from different $\mathscr{D}_s$ by Bayesian model averaging[39].

Given a metric-speed dataset from one $\mathscr{D}_s$, we used Bayesian linear regression (BLR) to fit the linear relationship between a metric and speed[38]. The benefit of BLR over ordinary least squares is that BLR naturally provides a way to set the regularization parameter (by setting the prior) and offers the posterior distribution of inferred parameters (e.g., slope), thus enabling a pure Bayesian analysis of the data. We follow the implementation of BLR in Bishop 2006[38].

In BLR, the relationship between metric and speed is modeled as

$$t = y(\mathbf{x}, \mathbf{w}) + \epsilon \tag{2}$$

where $\epsilon \sim \mathcal{N}\left(\epsilon | 0, \beta^{-1}\right)$, $\beta$ is a scalar representing precision, and $y = \mathbf{w}^T \mathbf{x}$. This equation indicates that the conditional distribution $p(t|\mathbf{x}, \mathbf{w})$ is a Gaussian distribution with mean $y$ and variance $\beta^{-1}$. The prior for parameter $\mathbf{w}$ is $\mathbf{w} \sim \mathcal{N}\left(\mathbf{w}|0, \alpha^{-1}\mathbb{I}\right)$ where $\mathbb{I}$ is a $2 \times 2$ identity matrix, and $\alpha$ is a scalar. Given the prior and conditional distribution, we can derive the posterior distribution $p(\mathbf{w}|\mathbf{t}, X)$ and predictive distribution $p\left(t_q|\mathbf{x}_q, \mathbf{t}, X\right)$, where $\mathbf{x}_q$ is the query label, $\mathbf{t}$ and $X$ are the data points in the metric-speed dataset, $t_q$ is the prediction. Both posterior and predictive distributions are Gaussian distributions. Hyperparameters $\alpha$ and $\beta$ were estimated by maximizing the marginal likelihood $p(\mathbf{t}|\alpha, \beta, X)$ through an iterative method[38].

Overall, given a metric-speed dataset (obtained from one sampled dataset $\mathscr{D}_s$), BLR provides the posterior distribution $p(\mathbf{w}|\mathbf{t}, X)$ and predictive distribution $p\left(t_q|\mathbf{x}_q, \mathbf{t}, X\right)$. To simplify the notation, the two distributions are written as $p(\mathbf{w}|\mathscr{D}_s)$ and $p(t_q|\mathbf{x}_q, \mathscr{D}_s)$, which follow $\mathcal{N}\left(\mathbf{w}|\mathbf{m}_{w,s}, \Sigma_{w,s}\right)$ and $\mathcal{N}\left(t_q|m_{t,s}, \Sigma_{t,s}\right)$, respectively.

We aggregate results from sampled datasets $\mathscr{D}_s$, i.e., $p(\mathbf{w}|\mathscr{D})$ and $p\left(t_q|\mathbf{x}_q, \mathscr{D}\right)$, by Bayesian model averaging[39]. Since each $\mathscr{D}_s$ is a random subsampling (under the restriction of an equal number of data points in each speed bin, see Methods: Data Preprocessing) of the $\mathscr{D}$, $p(\mathbf{w}|\mathscr{D}) = \sum_s p(\mathbf{w}|\mathscr{D}_s)p(\mathscr{D}_s|\mathscr{D}) = \sum_s p(\mathbf{w}|\mathscr{D}_s)/B$ where $B = 50$ is the total number of sampled datasets. $p(\mathbf{w}|\mathscr{D})$ is a mixture of Gaussian distributions. For simplicity, we approximate it as a single Gaussian distribution (see details in SI Methods). The mean of the approximated $p(\mathbf{w}|\mathscr{D})$ is

$$\mathbf{m}_w = \frac{1}{B}\sum_s \mathbf{m}_{w,s} \tag{3}$$

The covariance is

$$\Sigma_w = \frac{1}{B}\sum_s \Sigma_{w,s} + \frac{1}{B}\sum_s (\mathbf{m}_{w,s} - \mathbf{m}_w)(\mathbf{m}_{w,s} - \mathbf{m}_w)^T \quad (4)$$

where the first term is the average of covariances, second term represents bias.

Similarly, $p(t_q|x_q, \mathscr{D})$ can be approximated by a Gaussian distribution with certain mean and covariance matrix (see Supplementary Methods). In fact, the mean and covariance matrix are in forms that, after a Gaussian approximation on $p(t_q|x_q, \mathscr{D})$, $t_q$ is still a linear function of $\mathbf{w}$, as can be explicitly written as below

$$t_q = \mathbf{w}^T \mathbf{x}_q + \boldsymbol{\epsilon} \quad (5)$$

where $\mathbf{w} \sim \mathcal{N}(\mathbf{w}|\mathbf{m}_w, \Sigma_w)$, $\boldsymbol{\epsilon} \sim \mathcal{N}(\boldsymbol{\epsilon}; 0, \sum_s \beta_s^{-1}/B)$, and $\beta_s$ is the best hyperparameter fitted using an iteration method from a sampled dataset $\mathscr{D}_s$ (see above). Overall, we obtained $p(\mathbf{w}|\mathscr{D})$ and $p(t_q|\mathbf{x}_q, \mathscr{D})$, which are both Gaussian distributions. This overall method pipeline is called Bayesian linear ensemble averaging (BLEA) in this paper. Mathematical details can be found in Supplementary Methods.

$p(\mathbf{w}|\mathscr{D})$ and $p(t_q|\mathbf{x}_q, \mathscr{D})$ allow us to estimate the confidence interval (CI) and assess statistical significance from Bayesian framework[40]. First, since the predictive distribution $p(t_q|\mathbf{x}_q, \mathscr{D})$ is a Gaussian, the 95% CI of the prediction (two-tailed) is given by an interval [a, b] such that $\Phi((a-\mu)/\sigma) = 0.025$ and $\Phi((b-\mu)/\sigma) = 0.975$, where $\Phi(\cdot)$ is a cumulative distribution function of a standard Gaussian distribution, $\mu$ and $\sigma$ are the predictive distribution mean and standard deviation. One example CI can be found in Fig. 1C, which well covers most of data points. 95% CI is also called a credible interval in the Bayesian framework[40].

Similarly, knowing the posterior distribution of slope $p(\mathbf{w}|\mathscr{D})$, we can also compute its 95% CI. For example, this is illustrated by the error bars in Fig. 1D.

We are interested in whether the slope fitted from $\mathscr{D}$ is statistically different from that fitted from the label-shuffled dataset (e.g., Fig. 1D). Therefore, we also prepared label-shuffled $\mathscr{D}_s$ from $\mathscr{D}$ (see Methods: Data Preprocessing), and ran the above analysis to obtain their label-shuffled posterior and predictive distributions. Both of the posterior distributions of the original and label-shuffled datasets are Gaussian, hence we defined the slope difference $d = w^{data} - w^{shuffle}$, which also follows a Gaussian distribution with a mean equal to the difference between the two slope means and a variance equal to the sum of the two variances. Based on the distribution of $d$, probability of direction, $p_d$, can be computed as the maximum of $P(d>0)$ and $P(d<0)$. $p_d$ represents the probability that $d$ to be positive or negative (depending on which is the most probable). It directly relates to $p$-value (from a frequentist framework, two-sided) by $p = 2 \times (1 - p_d)$, where the null hypothesis is that $d = 0$ and alternative hypothesis is that $d \neq 0$[40]. Thus, statistical statements can be made based on the $p$-values. One-side $p$ value is $p_d$ or $1 - p_d$ depending on the one-side direction.

We are also interested in whether the speed-averaged metric computed under the $\mathscr{D}$ is statistically different from that computed under the hypothetical independent firing grid cell assumption (IFGC, Fig. 6). For each $\mathscr{D}_s$, we averaged the metric value across speed values. This gives one $\bar{t}_s$. Fifty $\bar{t}_s$ were concatenated and fitted by a Gaussian distribution using the maximum log-likelihood method, as an approximation of $p(\bar{t}_s|\mathscr{D})$. Therefore, we can use the same method above to determine whether the speed-averaged metric computed under original dataset is statistically different from that computed from the IFGC (Fig. 6B−E).

## Bin average and Ledoit-Wolf estimator

We approximate the neural population responses (neural states for short) as a Gaussian distribution:

$$\mathbf{r}(\mathbf{x}) = \boldsymbol{\mu}(\mathbf{x}) + \mathcal{N}(\boldsymbol{\epsilon}; 0, \Sigma(\mathbf{x})) \quad (6)$$

where $\mathbf{r} \in \mathscr{R}^N$ represents a neural state containing $N$ neurons, and $\mathbf{x} \in \mathscr{R}^M$ represents $M$ labels. Labels are defined broadly. They can be stimulus parameters (e.g., grating image orientation, object positions), an agent's latent state (e.g., latent dynamics factor, emotion), or an agent's behavioral labels (e.g., agent speed, agent position). $\boldsymbol{\mu}$ is the mean of neural state, modeled as a continuous function of the labels. $\boldsymbol{\mu}$ is also referred to as a manifold. $\boldsymbol{\epsilon}$ is white noise with a covariance $\Sigma(\mathbf{x})$. Given noisy neural states $\mathbf{r}$ and corresponding labels $\mathbf{x}$, our goal is to infer the smoothly varying manifold $\boldsymbol{\mu}$ and covariance $\Sigma$.

Bin averaging is a straightforward estimation method. This approach divides the entire range of label $\mathbf{x}$ into small bins. Data points $\mathbf{r}_i$ within each bin are considered to have the same label $\mathbf{x}_i$. Hence, the manifold can be estimated by sample average $\boldsymbol{\mu}(\mathbf{x}_i) = \langle \mathbf{r} \rangle_{\mathbf{x}_i}$, where $\langle \cdot \rangle_{\mathbf{x}_i}$ denotes averaging over the data points within bin $\mathbf{x}_i$. Similarly, the covariance $\Sigma$ can be estimated by the sample covariance matrix.

However, when the number of data points in each small bin is sparse and the neural state dimensionality is high (i.e., a large number of recorded neurons), bin averaging can lead to unreliable−and sometimes even non-invertible−estimation of the covariance matrix[24]. To address this, the shrinkage method was proposed. This method is equivalent to adding L2 regularization to the maximum likelihood estimation of the covariance matrix, guiding the estimation towards a more structured assumption (e.g., an identity matrix)[43]. In particular, this paper uses:

$$\Sigma = (1 - \lambda)S + \lambda \frac{\text{Tr}(S)}{N} \mathbb{I} \quad (7)$$

where $S$ is the sample covariance, $\lambda$ is the shrinkage coefficient estimated by the Ledoit-Wolf (LW) shrinkage algorithm[43], $N$ is the number of neurons, and $\mathbb{I}$ is an identity matrix. This algorithm is implemented by a Python function 'sklearn.covariance.LedoitWolf'.

## Gaussian process with Kernel Regression

One disadvantage of the bin average and LW methods is that the estimation of one bin's covariance does not use data from adjacent bins. Ideally, the manifold and covariance matrix are smooth over label values. Data in adjacent bins provide certain information about the current bin. Therefore, we developed the Gaussian Process with Kernel Regression (GKR) method to infer smoothly varying manifold and covariance from noisy neural states. GKR has two major steps: (1) inferring the manifold via a Gaussian process, and (2) inferring the covariance matrix. Note that while Gaussian processes have been used in previous studies to infer the firing of individual grid cells[76,77], our method, GKR, which is partially based on Gaussian processes, focuses on cell-to-cell statistics.

In step 1, each component of $\mathbf{r}$ across all time bins is standardized to have a mean of zero and a variance of one. Denoting the standardized $\mathbf{r}$ as $\tilde{\mathbf{r}}$. $\tilde{\mathbf{r}}$ then is modeled as $\tilde{\boldsymbol{\mu}} + \beta^2 \boldsymbol{\eta}$, where $\boldsymbol{\eta}$ is a standard Gaussian noise, and $\beta$ is a scalar parameter. The manifold $\tilde{\boldsymbol{\mu}}$ is modeled as an N-independent Gaussian process written as $\tilde{\boldsymbol{\mu}} \sim \mathscr{GP}^N(0, k_\mu)$, i.e., with zero mean and a kernel function $k_\mu : \mathscr{R}^M \times \mathscr{R}^M \to \mathscr{R}$ to control the "closeness" of $\tilde{\boldsymbol{\mu}}$ given two different labels $\mathbf{x}$[38]. A shared kernel for all components of $\tilde{\boldsymbol{\mu}}$ is used in this paper. Although the kernel is shared by all components $\tilde{\boldsymbol{\mu}}$, it has different parameters for different components of the label, i.e., $k_\mu(\mathbf{x}, \mathbf{x}') = \prod_{i=1}^M k(x_i, x_i') + c$, where $x_i$ is the $i$ th component of a label $\mathbf{x}$, and $c$ is a constant parameter. The kernel for $x_i$ is

$$k(x_i, x_i') = \sigma_i^2 \exp\left(-\frac{(x_i - x_i')^2}{2l_i^2}\right) \quad (8)$$

where $\sigma_i$ and $l_i$ are parameters. If the $x_i$ is a circular variable, a sine wrapping is applied:

$$k(x_i, x_i') = \sigma^2 \exp\left(-\frac{\sin^2(2\pi(x_i - x_i')/p_i)}{2l_i^2}\right) \quad (9)$$

where $p_i$ represents the period of the circular variable. Based on all these modeling, the problem of inferring $\widetilde{\boldsymbol{\mu}}$ from noisy data $\widetilde{\mathbf{r}}, \mathbf{x}$ becomes a classical Gaussian process regression problem, where the parameters $\{\beta, l_i, \sigma_i, c_i\}$ are optimized to maximize the log-likelihood of a joint Gaussian distribution for $\widetilde{\boldsymbol{\mu}} \sim \mathcal{GP}^{\mathcal{N}}\left(0, k_\mu\right)$. Finally, $\widetilde{\boldsymbol{\mu}}$ is unstandardized back to $\boldsymbol{\mu}$.

In many scenarios, the label $\mathbf{x}$ spans a large continuous range rather than a few discretized values (e.g., possible positions of a rat in a navigation task). In this case, Gaussian process regression requires computing a large kernel matrix, leading to expensive matrix manipulations[78]. To reduce this, we employed a variational inducing variable method[78]. It approximates training label values with a smaller set of inducing points $\mathbf{z}$, thereby reducing the time complexity. In this paper, inducing points were initialized as a randomly sampled subset of the original training labels (200 inducing points), and were optimized during the optimization of Gaussian process regression. Gaussian process regression with inducing variables method is implemented in the Python GPflow package[79].

The above step one infers the manifold $\boldsymbol{\mu}(\mathbf{x})$. Step two infers the covariance matrix $\Sigma(\mathbf{x})$. Define the gram matrix of a point $(\mathbf{r}_i, \mathbf{x}_i)$ as $C(\mathbf{x}_i)(\mathbf{r}_i - \boldsymbol{\mu}(\mathbf{x}_i))(\mathbf{r}_i - \boldsymbol{\mu}(\mathbf{x}_i))^T$, we estimate the covariance matrix at $\mathbf{x}$ as

$$\Sigma(\mathbf{x}) = \sum_i k_L(\mathbf{x}, \mathbf{x}_i) C(\mathbf{x}_i) + \eta \mathbb{I} \quad (10)$$

where $i$ sums over all training data points, and $\eta = 10^{-6}$ is a small number for numerical stability (keeping the covariance invertible even in the first term is small). $k_L(\mathbf{x}, \mathbf{x}_i)$ is a weight kernel that represents the contribution of $C(\mathbf{x}_i)$ in estimating the covariance matrix at label $\mathbf{x}$. It is normalized such that $\sum_i k_L(\mathbf{x}, \mathbf{x}_i) = 1$. To gain an intuition of this method, consider a simple case where (up to a normalization) $k_L(\mathbf{x}, \mathbf{x}_i) = 1$ if $||x_i - x|| < \delta$ and zero otherwise, step two is simply a sample covariance in a small bin of half-width $\delta$.

Since we assumed covariance is a smooth function over $\mathbf{x}$, $C(\mathbf{x}_i)$ of adjacent $\mathbf{x}_i$ should still contribute to the estimation of $\Sigma(\mathbf{x})$. Therefore, we used a gradually decaying weight kernel

$$k_L(\mathbf{x}, \mathbf{x}') = \kappa \exp\left(-0.5(\mathbf{x}-\mathbf{x}')^T LL^T (\mathbf{x}-\mathbf{x}')\right) \quad (11)$$

with $\kappa$ as a normalization factor ensuring $\sum_i k_L(\mathbf{x}, \mathbf{x}_i) = 1$, and $L$ is an $M \times M$ upper triangular matrix, interpreted as the Cholesky decomposition of a semi-positive definite precision matrix $LL^T$. Note that the precision matrix has non-diagonal terms, hence the interactions between different label components are considered.

Parameter $L$ is optimized to maximize the Gaussian log-likelihood of the data

$$\mathcal{L}(L) = -\sum_j \left[\log|\Sigma(x_j)| - (\mathbf{r}_j - \boldsymbol{\mu}(\mathbf{x}_j))^T \Sigma^{-1}(\mathbf{x}_j)(\mathbf{r}_j - \boldsymbol{\mu}(\mathbf{x}_j))\right] \quad (12)$$

where terms irrelevant to covariance are omitted. Notably, while $\Sigma$ is the weighted average of the Gram matrices from the training set, the log-likelihood function $\mathcal{L}$ should be evaluated from the validation set, where we used different indices $i, j$ to distinguish (Eq. 10 using training set). Setting the log-likelihood function on the training set would result in $\Sigma(\mathbf{x})$ converging to the Gram matrix $C(\mathbf{x})$. This can be demonstrated by computing $\Sigma(\mathbf{x})$ to satisfy the condition $\partial \mathcal{L}/\partial \Sigma = 0$. Therefore, splitting between training for computing covariance and validation for computing likelihood is necessary.

Overall, we use the following procedure to fit the manifold and covariance from a dataset. In step one, the entire train dataset was used for Gaussian process regression, obtaining a continuous manifold function $\boldsymbol{\mu}$. In step two, the train dataset was split into batches, each containing 3000 data points (except for the final batch). Each batch was further split into train and validation sets (0.66:0.33). The train set was used for computing the covariance matrix given an $L$ (initialized as an identity matrix), and the validation set was used to compute the log-likelihood function. The log-likelihood was maximized by an Adam optimizer (gradient applied on $L$). This batch training was repeated for 30 epochs. Finally, with the optimized $L$, the whole dataset was used for computing covariance (Eq. 10). The computer code for implementing GKR is provided at https://github.com/AgeYY/speed_grid_cell_information.git.

### Testing the normality assumption in data

We are interested in whether $p(\mathbf{r}|\mathbf{x})$ follows a normal distribution, as assumed by GKR. We first inspected the case of an example cube that centered at (11 cm, 37 cm, 18 cm/s) with edge lengths of (10 cm, 10 cm, 10 cm/s) in the label space. Data (from a $\mathscr{D}_s$ sampled from R1M2) within the cube were collected and projected into their PC1-PC2 plane, as well as PC1 axis and PC2 axis for visualization. Using the projected data, we fitted the optimal 2D and 1D normal distributions using maximum likelihood estimation and overlapped the optimal normal distributions with the projected data for direct visual comparisons. To formally assess normality, we performed Henze−Zirkler test to the projected 2D data, and Shapiro−Wilk tests to the projected 1D data (see Supplementary Fig. 10). This example cubic data shown in Supplementary Fig. 10 does not follow a normal distribution.

Next, we examined normality across sampled cubes. For each $\mathscr{D}_s$ (one $\mathscr{D}_s$ was sampled for one experimental configuration, e.g., R1M1, R1M2 etc.), we sampled 300 cubes. Data within each cube were projected onto their first 20 PCs. For each cube, we randomly sampled one PC, and applied Shapiro−Wilk test to assess normality. A $p$-value below 0.05 was considered indicative of non-normality. The percentage of non-normal cubes is shown in Supplementary Fig. 10. We performed normality tests on the projected PC rather than the full high-dimensional space, as testing in the full high-dimensional space would require significantly more data and could lead to unreliable results.

### Gaussian process regression with kernel sampling (GKR-S)

GKR assumes the conditional probability $p(\mathbf{r}|\mathbf{x})$ follows a normal distribution, whereas GKR-S employs a non-parametric approach that does not impose this assumption. The first step of GKR-S is identical to GKR, using Gaussian process to infer the mean $\boldsymbol{\mu}(\mathbf{x})$. We then computed the residue as $\boldsymbol{\epsilon}(\mathbf{x}_i) = \mathbf{r}_i - \boldsymbol{\mu}(\mathbf{x}_i)$, where the subscript $i$ denotes a training data point. The residual vector $\boldsymbol{\epsilon}(\mathbf{x}_i)$ is also referred to as $\boldsymbol{\epsilon}_i$.

The goal is to infer the residue distribution $p(\boldsymbol{\epsilon}|\mathbf{x})$ (which was assumed to be Gaussian in GKR). To achieve this, we employ a resampling strategy. We assign a weight to each training data point, where points closer to query label $\mathbf{x}_q$ receive higher weights. In this study, we use a Gaussian kernel to determine these weights

$$k(\mathbf{x}, \mathbf{x}') = \kappa \exp\left(-0.5(\mathbf{x}-\mathbf{x}')^T \Sigma^{-1}(\mathbf{x}-\mathbf{x}')\right) \quad (13)$$

with $\kappa$ as a normalization factor ensuring $\sum_i k(\mathbf{x}, \mathbf{x}_i) = 1$, and for simplicity, $\Sigma$ is a diagonal matrix. We determined the diagonal elements $\Sigma_{jj}$ heuristically using Silverman's rule-of-thumb[80]: $\sqrt{\Sigma_{jj}} = \left(\frac{4}{d+1}\right)^{\frac{1}{d+4}} n^{\frac{-1}{d+4}} \sigma_j$ where $\sigma_j$ is the standard deviation of the $j$-th label, $n$ is the number of training data, and $d$ is the dimensionality of the label $\mathbf{x}$.

For a given query label $\mathbf{x}_q$, this kernel function assigns a weight for each training data point. These weights serve as priors for resampling the training data. The sampled data are thought to be drawn from the

conditional distribution $p(\boldsymbol{\epsilon}|\mathbf{x})$. Consequently, the statistics of $p(\boldsymbol{\epsilon}|\mathbf{x})$ and $p(\mathbf{r}|\mathbf{x})$ can be empirically computed.

If one is interested only in the mean and covariance of $p(\mathbf{r}|\mathbf{x})$, they can be computed analytically without sampling data. Under the assumption of an infinite amount of resampling, the distribution $p(\boldsymbol{\epsilon}|\mathbf{x})$ is equivalent to

$$p(\boldsymbol{\epsilon}|\boldsymbol{x}) = \sum_i \delta(\boldsymbol{\epsilon} - \boldsymbol{\epsilon}_i) k(\mathbf{x}_i, \mathbf{x}) \qquad (14)$$

where $\delta(\cdot)$ is the Dirac delta function. Consequently, the mean of $p(\mathbf{r}|\mathbf{x})$ can be computed analytically as $\boldsymbol{\mu}(\mathbf{x}) + \sum_i k(\mathbf{x}_i, \mathbf{x})\boldsymbol{\epsilon}_i$.

When estimating the covariance, for simplicity, we assume that $\boldsymbol{\mu}(\mathbf{x})$ is a close estimation of the true mean, then $\sum_i k(\mathbf{x}_i, \mathbf{x})\boldsymbol{\epsilon}_i \approx 0$. This significantly simplifies the expression of noise covariance to $\sum_i k(\mathbf{x}_i, \mathbf{x})\boldsymbol{\epsilon}_i\boldsymbol{\epsilon}_i^\mathsf{T}$. For computational efficiency, we dropped training data points for which $k(\mathbf{x}_i, \mathbf{x}) < 10^{-6}$. It is worth noting that this mathematical form of covariance estimation is similar to the one used in GKR (Eq. 10), implying that the second step of the GKR may be thought as a case of a resampling method.

Using the above formula, we obtained both the mean and noise covariance, that allow further computation of other quantities as described in Methods: Computing Geometric Properties of the Speed-Slice Manifold at Different Speeds. In particular, the projected noise is estimated using Eq. (21), which is mathematically equivalent to resampling an infinite amount of data, projecting it onto the tangent plane, and then computing the projected data's covariance matrix. This paper focuses on the linear Fisher information which sets a fundamental limit on the variance of any unbiased linear estimator[81]. Both projected noise and linear Fisher information (referred to simply as Fisher information in this paper) are fully determined by the mean and covariance matrix.

## One-dimensional and two-dimensional synthetic datasets

Synthetic datasets are modeled as Gaussian distributions

$$\mathbf{r}(\mathbf{x}) = \boldsymbol{\mu}(\mathbf{x}) + \mathcal{N}(0, \Sigma(\mathbf{x})) \qquad (15)$$

The covariance matrix is $\Sigma(\mathbf{x}) = LL^\mathsf{T}$ where

$$L_{ij}(\mathbf{x}) = (\alpha\mu_i(\mathbf{x}) + \nu) \cdot e^{-\gamma|i-j|} \qquad (16)$$

with $\nu$ representing a constant for stimuli-independent noise, and $\gamma$ as the non-diagonal decay rate. Note that the covariance matrix $\Sigma$ depends on the manifold $\boldsymbol{\mu}$.

In the one-dimensional synthetic model, $i$ th component of the manifold $\boldsymbol{\mu}$ is given by

$$\mu_i(x) = g_i \frac{\mathcal{VM}(x - z_i, 1/\sigma^2)}{\mathcal{VM}(0, 1/\sigma^2)}, \qquad (17)$$

where $\mathcal{VM}(\cdot)$ denotes a von Mises function, $g_i \sim U(0.5, 1.5)$ is a random gain, $z_i = 2i\pi/N$ is the preferred label value for the $i$-th neuron, and $\sigma = 0.3$ is the tuning width. $x$ is a circular scalar label ranging from 0 to $2\pi$. Parameters for generating covariance matrix (Eq. 16) are: $\alpha = 0.2, \nu = 0.05, \gamma = 1$.

In the two-dimensional synthetic model, the $i$ th component of manifold $\boldsymbol{\mu}$ is

$$\mu_i(\mathbf{x}) = \exp\left(-\frac{||\mathbf{x} - \mathbf{z}_i||^2}{2(\sigma\lambda_i)^2}\right), \qquad (18)$$

where $\mathbf{z}_i \sim U([-1,1], [-1,1])$ is the center of the receptive field of neuron $i$, $\sigma = 0.3$, and $\lambda_i \sim U(0.5, 1.5)$ controls the tuning width. The label $\boldsymbol{x}$ is two-dimensional, with each component ranging from $-1$ to 1. Parameters for generating covariance matrix (Eq. 16) are: $\alpha = 0.5, \nu = 0.1, \gamma = 1$.

To generate a synthetic dataset of size $T$, $T$ labels $\mathbf{x}$ were uniformly sampled from the entire range. Each sampled label $\mathbf{x}$ was then used to compute one manifold point $\boldsymbol{\mu}$ and one covariance matrix $\Sigma$, thus generating one $\mathbf{r}$ using a Gaussian distribution (Eq. 15). $T$ labels generate $T$ data points.

When visualizing the ground truth of synthetic datasets in Fig. 2B, 100 labels $\mathbf{x}$ were randomly sampled. Then manifold points $\boldsymbol{\mu}$ and covariance matrices were computed. Manifold points were then fed into a PCA, dimensionally reduced to the first 2/3 dimensions (two for Fig. 2 and three for Supplementary Fig. 4). The covariance matrices were also projected onto the PCA subspace, transforming to a $2 \times 2/3 \times 3$ matrix. The eigenvalues of this $2 \times 2/3 \times 3$ matrix were visualized as the lengths of the ellipsoid's major axes (Fig. 2, Supplementary Fig. 4); and eigenvectors were visualized as the ellipsoid's major axes directions.

## Computing the relative prediction error of a metric to the ground truth in the synthetic dataset

We evaluated the performances of estimators (Bin average, LW, GKR) by comparing their predictions to ground truth. We evaluated several metrics: (1) manifold $\boldsymbol{\mu}$ (2) covariance matrix $\Sigma$ (3) Riemannian metric $(\partial\boldsymbol{\mu}/\partial\mathbf{x})^\mathsf{T}(\partial\boldsymbol{\mu}/\partial\mathbf{x})$ (4) Linear Fisher information $(\partial\boldsymbol{\mu}/\partial\mathbf{x})^\mathsf{T}\Sigma^{-1}(\partial\boldsymbol{\mu}/\partial\mathbf{x})$ (5) Precision matrix $\Sigma^{-1}$. $\partial\boldsymbol{\mu}/\partial\boldsymbol{x}$ was estimated numerically by finite difference.

For each condition (number of data points or number of neurons, Fig. 2), ten synthetic datasets were sampled. For each dataset, all data were used for training the estimator. Trained estimator predicts the values of metrics at other 100 randomly sampled labels. The relative estimation error is the mean of $\|M_i - \hat{M}_i\|_F / \|M_i\|_F$ over all 100 label $i$, where $M_i$ is the ground truth quantity while $\hat{M}_i$ is the estimated quantity, $\|\cdot\|_F$ is the Frobenius norm.

## Fit and visualize grid cell population manifold

GKR was applied to $\mathscr{D}_s$ to fit a manifold and a covariance matrix. Since $\mathscr{D}_s$ has three labels (two for locations and one is the speed), the full manifold is an intrinsically three-dimensional object. For the ease of visualization, we visualized slices of the manifold instead.

First, we visualized the manifold representing different locations but fixing the speed at 20 cm/s, i.e., speed-slice manifold (SSM). A 30 by 30 grid of positions was sampled from the entire OF space. Along with the fixed speed of 20 cm/s, these labels were fed into the fitted GKR to predict the manifold points on SSM. These 900 predictions were first reduced to 6 dimensions using PCA, then projected non-linearly into 3 dimensions using Uniform Manifold Approximation and Projection (UMAP), implemented by the Python umap-learn package. The parameters used were 'n_neighbors' = 100, 'min_dist' = 0.8, 'metric' = 'cosine', and 'init' = 'spectral'. To visualize the continuous manifolds, we interpolated small surfaces of adjacent x, y coordinate predictions using the plot_surface function in the Matplotlib Python package (Fig. 3B).

We visualized other manifold slices similarly. Figure 3D considered four adjacent spatial points (centered at $x = 75$ cm, $y = 75$ cm, with an adjacent points' distance of 2 cm) and varying speed values. Supplementary Fig. 7C considered four distant spatial points (centered at $x = 75$ cm, $y = 75$ cm, with an adjacent distance of 20 cm). Supplementary Fig. 7D considered a slice with a fixed $x = 75$ cm. PCA analysis on these slice manifolds suggested they are low-dimensional (3 PCs are sufficient to explain 80 percent of the variance). Hence, these manifold slices were directly visualized in the space of the first two/three principal components.

## Persistent Homology Barcode

Persistent homology is a method to analyze the topological structure of data clouds[82]. Each point in the data cloud was replaced by a small ball of radius $r$. If the distance of two points is smaller than $2r$, then

they would be connected. Roughly speaking, a graph with dots and connected lines is called a simplicial complex. Simplicial complex can have several holes of different dimensions (0D hole means a single component connecting all points; 1D hole is a loop; 2D hole is a cavity). As the dot ball radius increases from 0 to infinity, different dots would be connected, resulting in different simplicial complexes. During this process, some holes emerge while some holes die out. The birth and dead time of different holes can be collected and represented as bars. All bars of the same hole dimensions form a barcode for that dimension. Usually most bars are short, they likely represent noise, while long-life bars indicate non-trivial topological structure of the data cloud. The number of long-life bars in each dimension is counted as a Betti number, written in $\beta_i$. For example, a loop manifold should have one long-life zero-D hole, one one-D hole and no 2D holes. Hence the corresponding Betti number should be $(\beta_0, \beta_1, \beta_2) = (1, 1, 0)$. In particular, a torus should have Betti numbers $(\beta_0, \beta_1, \beta_2) = (1, 2, 1)$. We used the software package Ripser to compute the barcode, accompanied with approximated sparse filtrations to increase computational efficiency[83] (epsilon approximation constant = 0.2, see more detail in Ripser[84]). Intuitively, instead of computing the distance matrix of all points in the data cloud, approximated sparse filtrations discard balls which are completely covered by other balls (under certain $r$).

To build an objective procedure counting the number of long-life bars in the barcode, we defined a bar-length threshold to distinguish long-life bars. Here we defined the length threshold heuristically same as previous study[35]. A data point (e.g., a neural state) is a n-dimensional array where n is the number of grid cells. All data points form a m-by-n matrix, where m is the number of data points. We then randomly rolled (periodic boundary) each column of the matrix. This shuffled dataset was then fed into persistent homology, the maximum bar length was collected. This shuffling procedure repeated 20 times and obtained the final maximum bar length among 20 shuffling. This is the bar-length threshold.

Specifically, $\mathscr{D}_s$ was used to fit the GKR. When estimating the topological structure of the full three-dimensional manifold, 6,400 random labels were randomly sampled, and input to GKR to generate 6,400 manifold points. To simplify these data points, in accordance with Gardner et al. 2022[35], we firstly projected these data points into six PC subspace, and then used k-means to compute 1,200 cluster centers. These centers were then fed into Ripser, and Betti numbers were estimated from the above procedure. $(\beta_0, \beta_1, \beta_2) = (1, 2, 1)$ suggests a signature of possible toroidal-like topology.

When estimating the topological structure of SSM (Fig. 3B, speed = 20 cm/s), 30-by-30 grid locations were collected, fed into the GKR to make predictions. These 900 manifold points were then projected into the first six PC subspace, and then fed into Ripser to compute Betti numbers (there's no need to use k-means approximation in this case, because 900 data points is a good number to computationally handle, unlike the full-manifold case above).

## Computing Geometric Properties of speed-slice manifold at different speeds

$\mathscr{D}_s$ was used to fit the GKR. The fitted manifold is a function of $x, y$ locations and speed $v$. For each fixed speed value, we randomly sample 500 locations denoted as $(x_i, y_i)$.

The SSM center is the averaged 500 manifold points, denoted as $\mathbf{\mu}_c(v)$.

The SSM radius is the averaged Euclidian distance of these 500 random points to the SSM center

$$R = \sum_i \|\mathbf{\mu}(x_i, y_i, v) - \mathbf{\mu}_c(v)\|_F / N \quad (19)$$

Tangent vector $\partial\mathbf{\mu}/\partial x|_{(x_i, y_i, v)}$ measures how sensitive the neural population representation to the change of $x$ location[27]. Let $\mathbf{a}_i(v) \equiv \partial\mathbf{\mu}/\partial x|_{(x_i, y_i, v)}$, $\mathbf{b}_i(v) \equiv \partial\mathbf{\mu}/\partial y|_{(x_i, y_i, v)}$ and $a_i(v), b_i(v)$ as vector length respectively, lattice area at a point $(x_i, y_i, v)$ is the area formed by the two tangent vectors

$$
\begin{aligned}
A_i(v) &= a_i(v)b_i(v)\sin\theta \\
&= a_i(v)b_i(v)\sqrt{1 - \cos^2\theta} \\
&= \sqrt{a_i^2(v)b_i^2(v) - \left(\mathbf{a}_i(v) \cdot \mathbf{b}_i(v)\right)^2}
\end{aligned} \quad (20)
$$

The average of lattice area over 500 random points is the (averaged) lattice area of a SSM, shown in Fig. 3E.

Fisher information matrix is defined as $J^T \Sigma^{-1} J$, where $J$ is the Jacobian matrix in respect to spatial location. Total Fisher information is the trace of Fisher information matrix, averaged over all 500 random points.

To compute the projected noise, Jacobian matrix was normalized to $\hat{U}$, such that each column (tangent vector) has a unit length. Projected noise matrix is

$$\Sigma^{proj} = \hat{U}^T \Sigma \hat{U} \quad (21)$$

Project noise is the trace of projected noise matrix, averaged across 500 randomly sample points.

## Independent Poisson Speed-Gain (IPSG) grid cells

We present an Independent Poisson Speed-Gain (IPSG) model of the grid cells, characterized by three key assumptions: (1) the grid cells fire independently; (2) grid cells exhibit Poisson firing; (3) running speed modulates the spatial rate map through an increasing gain factor. Mathematically, the firing of a grid cell $i$ is given by:

$$r_i(x, v) \sim \text{Poisson}\left(f_i(v)M_i(\mathbf{x})\right) \quad (22)$$

where $x$ represents the 2D position, $f_i(v)$ denotes the speed-dependent gain, which is a monotonically increasing function. Substituting this into Eq. (19), we obtain the manifold size:

$$R = \frac{1}{A}\int_{\mathscr{X}} \|\mathbf{\mu}(x, v) - \bar{\mathbf{\mu}}(v)\| dx = \frac{1}{A}\int_{\mathscr{X}} \sqrt{\sum_{i=1}^n f_i^2(v)(M_i(\mathbf{x}) - \bar{M}_i)^2} dx, \quad (23)$$

where $A$ is the area of the open field $\mathscr{X}$; $\bar{M} = \int_{\mathscr{X}} M(x)dx/A$; $\mathbf{\mu}$ is the vector of mean firing rates with $\mathbf{\mu}_i(\mathbf{x}, v) = f_i(v)M_i(\mathbf{x})$, and $\bar{\mathbf{\mu}}(v)$ is its spatial average over $\mathscr{X}$. Therefore, Eq. (23) states that the manifold size increases with running speed.

To evaluate the total noise, we approximate Poisson firing with a Gaussian firing, leading to a population noise covariance matrix that is a diagonal (because of the independent firing), with diagonal element $\Sigma_{ii} = \mu_i(\mathbf{x}, v)/\Delta t$ where $\Delta t$ denotes the time window over which the firing rate is measured. The total noise, defined as the trace of the covariance matrix and averaged over the space $\mathscr{X}$, is given by:

$$\sigma^2 = \sum_{i=1}^n f_i(v)\bar{M}_i/\Delta t, \quad (24)$$

which also increases with speed.

Finally, the (linear) Fisher information averaged of $\mathscr{X}$ is

$$I(v) = \frac{1}{A}\int_x \left(\frac{\partial\mathbf{\mu}}{\partial\mathbf{x}}\right)^T \Sigma^{-1} \left(\frac{\partial\mathbf{\mu}}{\partial\mathbf{x}}\right) dA = \frac{1}{A}\sum_{i=1}^n f_i(v)\int_x \frac{||\nabla M_i(\mathbf{x})||^2}{M_i(\mathbf{x})} dA, \quad (25)$$

which also increases with running speed.

 

## Simulation of IPSG grid cells

To further evaluate whether IPSG grid cells can reproduce the increase in manifold size, total noise and Fisher information with speed (Figs. 3, 4, and 5), we simulated simple IPSG grid cells. The rate map of each grid cell is $M_i(\mathbf{x}) = \sum_a \cos(\frac{2\pi}{L}k^a\mathbf{x} + \phi_i^a) + 4$, where $k^a$ ($a = 1, 2, 3$) represents unit vectors pointing at 0, 60, and 120 degrees, respectively. $L = \pi$ is the spatial period, with the simulated field spans $[-\pi, \pi] \times [-\pi, \pi]$. The phase offset $\phi_i^a$ was randomly sampled from a uniform distribution over $[0, 2\pi)$; The additive factor of 4 ensures positive firing. All grid cells share the same speed gain function $f_i(v) \equiv f(v) = 20 - 20\exp(-v/10)$, which models a saturating speed gain, as implied by Hinman et al. 2016[11].

We simulated 10 IPSG grid cells, and randomly sampled 10,000 $\mathbf{x}$ and $v$ from uniform distributions, with $\mathbf{x}$ values ranging from the whole field and speed values ranging from 0 to 40. For each sampled $\mathbf{x}$ and $v$, we computed the mean firing rate of each grid cell, and then generated Poisson spikes, resulting in a simulated dataset containing spike counts along with their corresponding $\mathbf{x}$ and $v$.

To estimate uncertainty, we performed bootstrap resampling on the dataset 10 times (with replacement), generating 10 resampled datasets. For each resampled dataset, we used GKR to fit the data and estimate the manifold size, total noise and Fisher information, as shown in Supplementary Fig. 12.

## Fisher information provides the upper bound of spatial classification accuracy

Consider a classification problem involving data from two small boxes centered at $\mathbf{x}_{\pm} = \mathbf{x}_c \pm \delta\mathbf{x}$. Denote the two classes as $\mathscr{C}_1$ and $\mathscr{C}_2$. In the process of evaluating SCA, we subsampled the data points so that two boxes have equal data set sizes. In line with this, the prior probabilities of a data point belonging to either class are equal, $p(\mathscr{C}_1) = p(\mathscr{C}_2) = 1/2$. We also assume the neural state $\mathbf{r}$ in box $i$ is approximately given by $\mathcal{N}(\mathbf{r}; \boldsymbol{\mu}_i, \Sigma)$, where $i$ can be 1 or 2. Here we derive the optimal classification accuracy if the classification boundary is linear (as used by the logistic classifier). A linear classification boundary means that the class is $\mathscr{C}_1$ if $y = w^T r - w_0 < 0$, and $\mathscr{C}_2$ otherwise.

Classification accuracy is the probability of a correct classification.

$$
\begin{aligned}
p(\text{correct}) &= p(y < 0, \mathscr{C} = \mathscr{C}_1) + (y \geq 0, \mathscr{C} = \mathscr{C}_2) \\
&= \frac{1}{2}\left[p(y < 0|\mathscr{C}_1) + p(y \geq 0|\mathscr{C}_2)\right] \\
&= \frac{1}{2}\left[p(\mathbf{w}^T\mathbf{r} < w_0|\mathscr{C}_1) + p(\mathbf{w}^T\mathbf{r} \geq w_0|\mathscr{C}_2)\right] \\
&= \frac{1}{2}\left[\Phi\left(\frac{w_0 - \boldsymbol{w}^T\boldsymbol{\mu}_1}{\sqrt{\boldsymbol{w}^T\Sigma\boldsymbol{w}}}\right) + 1 - \Phi\left(\frac{w_0 - \boldsymbol{w}^T\boldsymbol{\mu}_2}{\sqrt{\boldsymbol{w}^T\Sigma\boldsymbol{w}}}\right)\right]
\end{aligned} \tag{26}
$$

where $\Phi(\cdot)$ is the cumulative density function of a standard normal distribution. We used the fact that, if $p(\mathbf{r}|\mathscr{C}_i)$ is a Gaussian, $\mathbf{w}^T\mathbf{r}$ is also a Gaussian with mean $\mathbf{w}^T\boldsymbol{\mu}_i$ and variance $\mathbf{w}^T\Sigma\mathbf{w}$.

Next, we find the optimal $P(\text{correct})$. Let $\partial P(\text{correct})/\partial w_0 = 0$, we get $w_0 = \mathbf{w}^T(\boldsymbol{\mu}_1 + \boldsymbol{\mu}_2)/2$. Let $\partial P(\text{correct})/\partial w = 0$, we get an equation $\Delta\boldsymbol{\mu}(2\mathbf{w}^T\Sigma\mathbf{w}) = 2\Sigma\mathbf{w}(\mathbf{w}^T\Delta\boldsymbol{\mu})$ where $\Delta\boldsymbol{\mu} = \boldsymbol{\mu}_2 - \boldsymbol{\mu}_1$. This equation has a general solution $\mathbf{w} \propto \Sigma^{-1}\Delta\boldsymbol{\mu}$. Substituting this back into accuracy, we get the optimal accuracy of the two boxes is $\Phi\left(\sqrt{\Delta\boldsymbol{\mu}^T\Sigma^{-1}\Delta\boldsymbol{\mu}/2}\right)$.

In our case, boxes were chosen to be symmetric, hence $\boldsymbol{\mu}_1 = \boldsymbol{\mu}(\mathbf{x}_c - \delta\mathbf{x}) \approx \boldsymbol{\mu}(\mathbf{x}_c) - \nabla\boldsymbol{\mu}\delta\mathbf{x}$, and similarly $\boldsymbol{\mu}_2 = \boldsymbol{\mu}(\mathbf{x}_c + \delta\mathbf{x}) \approx \boldsymbol{\mu}(\mathbf{x}_c) + \nabla\boldsymbol{\mu}\delta\mathbf{x}$, where $\delta\mathbf{x} = \delta l\hat{\mathbf{e}}$. Therefore, $\Delta\boldsymbol{\mu} = 2\nabla\boldsymbol{\mu}\delta\mathbf{x}$. The upper bound of accuracy becomes $\Phi\left(\sqrt{\delta\mathbf{x}^T I\delta\mathbf{x}}\right)$ where $I$ is the Fisher information. In our numerical procedure, $\hat{\mathbf{e}}$ is a random unit vector. The overall upper bound of accuracy is given by the integration across all angles of $\hat{\mathbf{e}}$

$$
p(\text{correct})_{\text{optimal}} = \int_0^{2\pi} \Phi\left(\sqrt{\delta\mathbf{x}^T I\delta\mathbf{x}}\right)d\theta/(2\pi) \tag{27}
$$

The relationship between optimal accuracy and the total Fisher information becomes intuitive if we assume the Fisher information is unbiased in all directions, which, biologically speaking, means that a rat has no directional bias in an open field. Under this assumption, the integral is trivial because the function inside is independent of direction. Second, the Fisher information becomes proportional to the identity matrix $Tr(I)\mathbb{I}$. Without loss of generality, let $\delta\mathbf{x} = \delta l(1, 0)^T$. The optimal accuracy becomes (Taylor expanded around zero) $0.5 + \delta l\phi(0)\sqrt{Tr(I)}$ where $\phi(0) = 1/\sqrt{2\pi}$. Therefore, optimal accuracy asymptotically increases with the square root of the total Fisher information.

We used Monte Carlo method to estimate the integral in Eq. (27). Specifically, we sampled 2D vectors from a 2-dimensional standard Gaussian distribution, then rescaled the 2D vectors to have a length equal to $\delta l$, resulting the sampled $\delta\mathbf{x}$. This sampling is unbiased with respect to angle because the 2-dimensional standard Gaussian distribution is isometric. Given Fisher information $I$, the upper bound was estimated as the average of $\Phi\left(\sqrt{\delta\mathbf{x}^T I\delta\mathbf{x}}\right)$ across all $\delta\mathbf{x}$.

## Test upper bound of spatial classification accuracy on synthetic datasets and grid cell population responses

The upper bound (Eq. 27) is straightforward in a one-dimensional $\delta x$

$$
p(\text{correct})_{\text{optimal}} = \Phi\left(\delta l\sqrt{I}\right) \tag{28}
$$

We tested upper bounds in both 1D and 2D synthetic data sets. For each parameter configuration (number of neurons $N$, number of data points $K$, and noise level $\alpha$; the other parameters were fixed as described in Methods: One-Dimensional and Two-Dimensional Synthetic Datasets), we generated $K$ data points. SCA was computed as described in 'Methods: Spatial classification accuracy'. On the other side, the generated $K$ data points were also used for fitting GKR. Fitted GKR make predictions of Fisher information, which then converted to upper bound (Eq. 27). Finally, the ground truth Fisher information of the synthetic datasets was also used to compute the upper bound. Results are shown in Supplementary Fig. 9.

We also inspected the upper bounds on the Grid cell datasets. GKR provides predictions of Fisher information, which were then used to compute the upper bounds. The upper bounds and SCAs of the R1M2 were shown in Supplementary Fig. 8C and Fig. 5C for $\mathscr{D}_s$ and $\mathscr{D}_s^{(6)}$ (projection to six PCs, see Methods), respectively. Upper bound-speed/SCA-speed array has $50 \times 8 = 400$ data points (fifty sampling $\mathscr{D}_s/\mathscr{D}_s^{(6)}$ times 8 speed bins). To have a quantitative comparison between upper bounds and SCA, we computed the Pearson correlations between the two arrays, denoted as $r$. The $p$-value (two-sided) and confidence interval (via Fisher z-transform) can be computed accordingly via Python package stats.peason[85,86].

## A two-neuron example demonstrating the effects of noise correlation

We present a simple two-neuron system to illustrate that the neuronal noise correlation can have either beneficial or detrimental effects, depending on the geometry of the noise covariance.

Consider a two-neuron system encoding two classes. The mean firing rates of the two neurons under class 1 are $\boldsymbol{\mu}_1 = (1, 0)^T$, and under class 2 are $\boldsymbol{\mu}_2 = (0, 1)^T$. The neuron's firing noise follows a Gaussian distribution with a covariance matrix given by:

$$
\Sigma = \begin{bmatrix} 1 & \rho \\ \rho & 1 \end{bmatrix} \tag{29}
$$

where the value 1 represents the variance of individual neural noise, and $\rho$ denotes the noise correlation. For $\Sigma$ to be a valid (positive-definite) covariance matrix, it must satisfy $1 - \rho^2 > 0$. Given the mean firing rates and noise covariance, we can compute the Fisher information:

$$I = (\Delta\boldsymbol{\mu})^T \Sigma^{-1} (\Delta\boldsymbol{\mu}) = \frac{2}{1 - \rho} \qquad (30)$$

where $\Delta\boldsymbol{\mu} = \boldsymbol{\mu_2} - \boldsymbol{\mu_1}$ represents the vector difference between the class-conditional firing means. The baseline Fisher information, when there is no correlation ($\rho = 0$), is 2. Positive correlation ($\rho > 0$) increases the Fisher information, hence it is information-beneficial; while negative correlation ($\rho < 0$) reduces the Fisher information, hence it is information detrimental. In general, negative correlation is not necessarily detrimental to information—it depends on the geometric relationship between the noise covariance and the signal. In this simple example, positive noise "reshapes" the noise covariance so that more noise aligns with the classification decision boundary, whereas negative noise "reshapes" it such that more noise is orthogonal to the decision boundary (Supplementary Fig. 13).

To show these effects in simulation, we generated 1000 total samples (500 per class) under three different noise conditions (see Supplementary Fig. 13): (1) beneficial correlation $\rho = 0.8 > 0$: noise aligns with the decision boundary; (2) detrimental correlation $\rho = -0.8 < 0$: noise projects onto the signal axis $\boldsymbol{\Delta\mu}$; and (3) independent noise $\rho = 0$: no correlation between neurons. Note that the total noises (trace of the covariance matrix) of three conditions are the same. With these generated data, we trained a logistic regression classifier to distinguish two classes for each condition. The classification accuracy quantifies the "goodness of information coding". We also indicate the theoretical Fisher information value in Supplementary Fig. 13.

### Independent Firing Grid Cells
We investigated the effect of grid cell activity correlation by comparing results from the original dataset $\mathscr{D}_s$ to those from hypothetical independent firing grid cells (IFGC). A classic method for generating independent firing cells involves shuffling trials within the same condition. Specifically, each cell's firing profiles are randomly permuted across trials within a condition. This approach preserves single-cell firing statistics while disrupting cell-to-cell firing correlation. We adapted this method when computing SCA of the IFGC (Fig. 6E). Recall that the key idea of SCA is to compute the classification performance on data within two nearby (spatial) boxes. We treat data within each box as a single condition, where each data point (an N-dimensional vector of single-cell firing rates) represents one trial. We then randomly permute each cell's firing rate across all data points within the box, breaking the cell-to-cell correlation. The SCA of this "trial-shuffled" data is called the SCA of IFGC (Fig. 6E).

We also adapted this "trial-shuffling" idea to compute the geometric metrics (total noise, projected noise, and Fisher information) for IFGC. Specifically, after fitting $\mathscr{D}_s$, GKR can predict mean and covariance at a condition $\mathbf{x}$. Consider GKR as a generative model, it generates infinite data points under the same condition $\mathbf{x}$. If we applied the above "trial-shuffling" procedure on these data points, and recompute the mean and covariance matrix, the mean remains unchanged, while the covariance matrix retains only the diagonal components of the original covariance matrix, with all off-diagonal components set to zero. Therefore, IFGC's GKR is same as the original GKR except only having the diagonal covariance matrix. With IFGC's GKR, the geometric metrics can be computed as previously described.

We compared speed-averaged metrics obtained from the original datasets to those obtained from IFGC. Methods of computing speed-averaged metrics along with statistical analysis can be found in the Methods: Bayesian linear ensemble averaging and statistical testing section.

### Reporting summary
Further information on research design is available in the Nature Portfolio Reporting Summary linked to this article.

## Data availability
Source data are provided with this paper. The grid cell spiking dataset collected by Gardner et al. (2022)[35] and used in this study is available in Figshare with the identifier: https://doi.org/10.6084/m9.figshare.16764508.v6. Source data are provided with this paper.

## Code availability
Code for reproducing the analyses in this article is available at https://github.com/AgeYY/speed_grid_cell_information.git.

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

## Acknowledgements

The authors gratefully acknowledge Robert Wang for reviewing the manuscript and providing insightful comments; Dr. Richard J. Gardner for clarifying details regarding the publicly available grid cell spiking dataset[35]; and Dr. Haoran Li for testing our computer code. This work was supported by grants from Incubator for Transdisciplinary Futures: Toward a Synergy Between Artificial Intelligence and Neuroscience (RW).

## Author contributions

Conceptualization and writing: Z.Y. and R.W.; methodology and investigation: ZY.; supervision: RW.

## Competing interests

The authors declare no competing interests.
