## [Transparent Peer Review file · Nature Communications]

Speed modulations in grid cell information geometry

Corresponding Author: Mr Zeyuan Ye

Version 0:

Reviewer comments:

Reviewer #1

(Remarks to the Author)

Summary

In this work, the authors examine the population level activity of grid cells through a new method they develop here, Gaussian process with kernel regression (GPKR). GPKR infers both the manifold (low-dimensional structure) of the data as well as the noise covariance. Through GPKR, they examine the influence of running speed the grid cell code, including how running speed correlates with the noisiness of firing. They analyze a previously published dataset of simultaneously recorded grid cells from Gardner et al., 2022. Consistent with prior analyses on this dataset, they find that GPKR reveals a torus manifold. They find that running speed dilates this manifold, which increases the fidelity of the spatial code, and that running speed increases the noisiness of firing, which decreases encoding accuracy. However, when looking at the two effects combined, the encoding accuracy net increases with running speed. In addition, they use GPKR to examine the impact of noise correlations on the spatial code. They find that the noise correlations are generally detrimental, meaning that the noise lies along the manifold and thus could be 'confused' for real signal by a decoder.

Overall, the work presented was interesting and easy to follow. In terms of impact, the authors present both an interesting manifold-discovery technique suited for single trial data and scientific results on grid code coding. Although I am not an expert on the development of techniques like GPKR, what they present seems reasonable and, from a very high level, relatively clear (that they iteratively optimize the mean and the covariance structure). It is also great that they included a GitHub link so that the code/analyses can be explored and used by others. However, on the scientific results side, it is unclear to me how novel or surprising their presented results are. More detailed comments on that, and on a few other items, are detailed below.

Major comments

- Grid cell firing rates tend to increase with running speed. The authors also mention this in the discussion. It is not clear to me how much this effect explains the results they see, or how they interpret their results considering this fact. If we also assume that neurons are Poisson-like in that the variance scales with the mean, would these two facts completely explain both the dilation of the manifold and the increase in noise with running speed? In other words, if the authors were to simulate grid cells according to Poisson firing and assuming that firing rate increases with running speed, would they see the same effects? And if so, would the authors interpret their findings to visualize *how* running speed increases coding fidelity? Or, perhaps as another way to visualize or quantify the change in the noise with running speed? As I write this, I am wondering if the authors view their own paper as more of a technique paper, or a scientific results paper, or somewhere in between.

- Along somewhat similar lines, regarding the other major scientific result presented - that the noise correlations in grid cells lie along the manifold and thus are detrimental to the fidelity of the spatial code - are surprising given what we know about grid cells. In this case, much prior work has indicated that grid cell activity arises from connectivity between cells, and thus noise is most likely correlated between cells (and not independent). I'm not sure the fact that this is a strong prediction (that noise is correlated across neurons and detrimental) is an issue, but it's just a bit unclear in this paper if the authors view this as a novel finding or an illustration of their technique.

- This paper analyzes a previously published dataset, but the authors do not make this entirely clear from the outset - the language in the main text is vague as to where the data came from (although it indicates it's from Gardner in the methods). I

would suggest changing the wording so it's clear what is being analyzed.

- I went through the code on GitHub. I didn't run attempt to the code on the Gardner data, which I see must be downloaded from their site directly. I did, however, try to run the GKR example. Unfortunately, I wasn't able to install the environment as provided in env.yaml. I detailed my notes below - I think there are several places where the implementation can be more user friendly. It's possible that the failure to download was on me (I didn't have a ton of time to devote to troubleshooting), but it might be worth having another novel user try to download and run the code.
- It seems that the code as uploaded was developed on Windows and might be only compatible with Windows OS (e.g. the `add_python_path`). If true, it would be helpful if the authors stated that on the github main page. Also, it looks like this code needs Nvidia GPUs – this and any other hardware dependencies would be nice to state upfront.
- in the code '`add_python_path`', there are a lot of hardcoded paths for the authors' directories. I can't tell what some of these are – e.g. 'GridCellTorus' or 'disentangle.' I can't find what these paths correspond to.
- I had issues with installing the packages listed in env.yaml (`cuda==8.8` had to be set to just `cuda`, and `hickle==...` needed to be `hickle==...`). Eventually the code errored out at trying to build the wheel for `mpi4py`.
- A note regarding the data folder: in the readme, it seems that there is some confusion about what the variables in the uploaded data correspond to (e.g. the spike times, the variable `z`, etc). It might be worth verifying with the authors of the dataset any outstanding questions, just so that the authors can be confident that they are using the variables correctly.

Minor comments

- A question regarding GPKR - does the user have to set the number of label variables? In other words, does the user have to already know the intrinsic dimensionality of the neural population?
- The authors state that they create speed-balanced datasets prior to doing any analyses. I don't think this is an issue, but it would be nice if they could be clearer about what sort of issues in their analyses might have arisen if they didn't do this processing on the data. Were they concerned that speed-imbalanced data would generate a less interpretable manifold, especially if they didn't know a priori how speed would influence the shape of the manifold?
- There were quite a few typos and sentences that were not grammatically correct throughout the text. This might get corrected later in the review process, but I think it's worth doing a thorough read-through to eliminate these errors, since it will have to happen at some point.

(Remarks on code availability)

I did not go through and review all the code extensively, but rather I tried to run the GKR example. I wasn't able to get the code running unfortunately. I detailed the issues I had in the review above. The code does include a readme, although I think the readme could be edited to be more helpful.

Reviewer #2

(Remarks to the Author)

In this manuscript, Ye and Wessel address the important issue of how a rodent's velocity affects information encoding in the medial entorhinal cortex's grid cells, and also explores whether the "noise" dimensions are orthogonal or not.

The manuscript sells itself as a methods paper, touting the "Gaussian Kernel Regression" as novel. While I am not an expert on Gaussian process modeling, I think this claim is a stretch. From what I understand, the first step is to let the mean of the GP be zero, and then to fit the covariance matrix. This is a perfectly reasonable approach, of course. I suspect the details might differ, but this approach does bear some similarities to Rule et al and Papastathopoulos et al, which build upon other papers' approaches.

It might be useful to draw a more detailed comparison to these other papers in the Methods section.

The presentation in the manuscript is clear and detailed, but at times I had the impression that the authors wanted to throw the kitchen sink at the problem. For instance, what is the point of doing topological data analysis on the homology classes of the filtrations to get the "bar codes", but then to stop short of using cohomology to decode? If velocity is a new variable in addition to the spatial coordinates, then what is the topology of this space as a whole? The authors refer to manifold dilation, which would invoke concepts from differential geometry, but topology is only concerned with continuity (And I do have to object to the claim that a Betti sequence of (1,2,1) indicates a torus---it can be a torus, but Betti sequences from the abelianization of algebraic geometry do not map one-to-one to unique topologies)

My recommendation: rewrite the main text to focus solely on the questions of neuroscientific import (and leave the methods section as is) and perhaps drop the topological data analysis to simplify the manuscript.

The main result, namely that the spatial information increases with speed, has sufficient merit to carry the entire manuscript. I do not think the authors should stress the "methods paper" aspect.

(Remarks on code availability)

Reviewer #3

(Remarks to the Author)

This paper is well-written and investigates how running speed modulates the information coding of grid cells. Using experimental data from previous study in Moser's Lab, the authors used a novel method, GKR, to infer the manifold and noise covariance of neural data. They showed that increasing speed will increase spatial accuracy as well as noise covariance (that is shown to be detrimental). However, the overall effect is the improvement of spatial coding, as shown in the increase of Fisher Information. GKR, potentially coupled with dimension reduction techniques, will provide a useful tool to understand neural population coding.

Some comments are:

- * What is the implication of the results on the underlying model of grid cells?
- * Figure 1C: is the increase of speed accuracy caused by the more divided spatial positions when the speed is large?
- * Figure 1: Will the choice of decoder (other than a logistic regressor) affect the result?
- * Equation (1): Does the assumption of GKR always hold (i.e., the data always follows a Gaussian distribution)?
- * Figure 4C: is there any reason why S1M1 has negative slope?
- * I'm curious about how this paper would reveal the coding properties of hippocampal cells and how it's different from grid cell investigated here? (If authors can test it straightaway, that would be awesome; but it doesn't imply authors have to do it in this paper)
- * Line 346: "do not"

(Remarks on code availability)

Reviewer #4

(Remarks to the Author)

This study investigates how running speed affects spatial coding in grid cells. The researchers developed a novel method called Gaussian Process with Kernel Regression (GKR) to analyze simultaneously recorded grid cell population activity from an information geometry perspective. Their key findings show that:

1. Higher running speeds lead to better spatial coding accuracy
2. Running speed increases both the size of the grid cell manifold (geometric representation) and noise levels
3. The positive effect of manifold expansion outweighs the negative effect of increased noise
4. Cell-to-cell noise correlations are detrimental to information coding

Strengths:

1. Novel methodology (GKR) that allows analysis of high-dimensional neural data without requiring repeated trials
2. Comprehensive validation of the method using synthetic data
3. Rigorous mathematical framework and statistical analysis
4. Clear connection between theoretical predictions and experimental results

I have two main concerns:

1. Choice of labels:

- The method assumes we know the relevant variables (position, velocity) a priori
- It imposes a particular structure on how information is encoded
- Could miss other important variables or combinations of variables
- Assumes separability of these variables

Specific impacts on analysis:

- Could artificially separate variables that are actually encoded jointly
- Might miss nonlinear combinations of variables
- Risk of confirmation bias - finding structure we assume exists
- Could underestimate the dimensionality of the true representation

Possible mitigations:

- Compare different label choices using model selection
- Use nonparametric approaches to discover relevant variables
- Test for interactions between different labels
- Validate findings using behavioral relevance

2. Choice of noise model:

- Real neural noise may be non-Gaussian
- Could have heavy tails or be skewed
- May vary with the signal

- Could have complex temporal structure

Impact on analysis:

- Could misestimate uncertainty
- May not capture true noise correlations
- Risk of over/under-fitting
- Could affect estimates of Fisher information

Possible mitigations:

- Compare different noise models (e.g., Poisson, Log-normal)
- Use nonparametric bootstrapping approaches
- Test robustness to noise model assumptions
- Validate predictions using held-out data

Other weaknesses I would like to see discussed:

1. The study focuses primarily on mathematical/theoretical aspects, with limited discussion of biological implications
 - a. Are there experiments where manipulating running speed artificially (e.g., treadmill) could establish causality, e.g. Campbell 2021
 - b. How can the method generalize to other brain regions or behaviors when one has to assume labels
 - c. Grid cells are known to react to environmental change (e.g. Barry et al 2012) How can environmental complexity or novelty interact with speed effects

2. Limited exploration of the mechanisms behind speed-dependent improvements in spatial coding
 - A. Expand the discussion of biological mechanisms that might explain the speed-dependent effects

3. Include more detailed discussion of the limitations of the GKR method and potential failure cases

Minor:

In the first sentence you write grid cells were recorded, but this study only uses data from a previous study, please modify this sentence to make it immediately clear that you used data from this reference.

(Remarks on code availability)

Version 1:

Reviewer comments:

Reviewer #1

(Remarks to the Author)

I appreciate the authors' responses to my comments, especially the inclusion of the comparison Poisson-like grid cells with mean firing that scales with running speed. I just have a few remaining comments:

1. I find the inclusion of GKR-S (added in response to another reviewer) helpful, but I also found its introduction in the manuscript somewhat confusing. The authors state that their main method violates the assumption of normality, but then proceed to use it anyway, later demonstrating that it produces results similar to GKR-S. I think this can be easily fixed, e.g. by introducing GKR-S later and stating there that the main method (GKR) yields comparable results, which also justifies continuing to focus on it as the primary method.
2. Regarding the IPSPG model: it is interesting (if not surprising) that this simplified formulation reproduces many of the key results - up until the results when noise correlations are considered. The model is simplified, as the authors say, but the fact that many results can be reproduced under these simplifications suggests to me that certain complexities in the real data (e.g. nonlinear relationships between running speed and firing rate or non-Poisson firing) are not critical for the core results presented here. This is useful for understanding, as it helps isolate which features must be added to a simplified model to approximate real data, thereby clarifying the functional role of those features. I would thus possibly re-state the impact of this analysis in the paper. Alongside this, the IPSPG model, as the authors note, cannot capture the effects of noise correlations. I wonder if therefore, it can be viewed as a null model for those analyses, essentially replacing the IFGC model, which would simplify presentation.

(Remarks on code availability)

Reviewer #3

(Remarks to the Author)

Overall, the authors have addressed my feedback well, and I have one remaining:

When the authors discussed this part (R3.1 What is the implication of the results on the underlying model of grid cells?), I think the authors need to consider:

1. Normally, there are three types of models of grid cells: CAN (continuous attractor network, rate-based), OI (Oscillatory-interference) and self-organising (learning models).
2. Kropff et al. 2021 mentioned that theta frequency is controlled by acceleration (not speed). How to reconcile with this finding?

(Remarks on code availability)

Reviewer #4

(Remarks to the Author)

I want to thank the authors for their detailed answers. I have no more comments or questions

(Remarks on code availability)

Reviewer #1 (Remarks to the Author):

Summary

In this work, the authors examine the population level activity of grid cells through a new method they develop here, Gaussian process with kernel regression (GPKR). GPKR infers both the manifold (low-dimensional structure) of the data as well as the noise covariance. Through GPKR, they examine the influence of running speed the grid cell code, including how running speed correlates with the noisiness of firing. They analyze a previously published dataset of simultaneously recorded grid cells from Gardner et al., 2022. Consistent with prior analyses on this dataset, they find that GPKR reveals a torus manifold. They find that running speed dilates this manifold, which increases the fidelity of the spatial code, and that running speed increases the noisiness of firing, which decreases encoding accuracy. However, when looking at the two effects combined, the encoding accuracy net increases with running speed. In addition, they use GPKR to examine the impact of noise correlations on the spatial code. They find that the noise correlations are generally detrimental, meaning that the noise lies along the manifold and thus could be 'confused' for real signal by a decoder.

Overall, the work presented was interesting and easy to follow. In terms of impact, the authors present both an interesting manifold-discovery technique suited for single trial data and scientific results on grid code coding. Although I am not an expert on the development of techniques like GPKR, what they present seems reasonable and, from a very high level, relatively clear (that they iteratively optimize the mean and the covariance structure). It is also great that they included a GitHub link so that the code/analyses can be explored and used by others. However, on the scientific results side, it is unclear to me how novel or surprising their presented results are. More detailed comments on that, and on a few other items, are detailed below.

Major comments

- Grid cell firing rates tend to increase with running speed. The authors also mention this in the discussion. It is not clear to me how much this effect explains the results they see, or how they interpret their results considering this fact. If we also assume that neurons are Poisson-like in that the variance scales with the mean, would these two facts completely explain both the dilation of the manifold and the increase in noise with running speed? In other words, if the authors were to simulate grid cells according to Poisson firing and assuming that firing rate increases with running speed, would they see the same effects? And if so, would the authors interpret their findings to visualize *how* running speed increases coding fidelity? Or, perhaps as another way to visualize or quantify the change in the noise with running speed? As I write this, I am wondering if the authors view their own paper as more of a technique paper, or a scientific results paper, or somewhere in between.

- Along somewhat similar lines, regarding the other major scientific result presented - that the noise correlations in grid cells lie along the manifold and thus are detrimental to the fidelity of the spatial code - are surprising given what we know about grid cells. In this case, much prior work has indicated that grid cell activity arises from connectivity between cells, and thus noise is most likely correlated between cells (and not independent). I'm not sure the fact that this is a strong prediction (that noise is correlated across neurons and detrimental) is an issue, but it's just a bit unclear in this paper if the authors view this as a novel finding or an illustration of their technique.

- This paper analyzes a previously published dataset, but the authors do not make this entirely clear from the outset - the language in the main text is vague as to where the data came from (although it indicates it's from Gardner in the methods). I would suggest changing the wording so it's clear what is being analyzed.

- I went through the code on GitHub. I didn't run attempt to the code on the Gardner data, which I see must be downloaded from their site directly. I did, however, try to run the GKR example.

Unfortunately, I wasn't able to install the environment as provided in env.yaml. I detailed my notes below - I think there are several places where the implementation can be more user friendly. It's possible that the failure to download was on me (I didn't have a ton of time to devote to troubleshooting), but it might be worth having another novel user try to download and run the code.

- It seems that the code as uploaded was developed on Windows and might be only compatible with Windows OS (e.g. the add_python_path). If true, it would be helpful if the authors stated that on the github main page. Also, it looks like this code needs Nvidia GPUs - this and any other hardware dependencies would be nice to state upfront.

- in the code 'add_python_path', there are a lot of hardcoded paths for the authors' directories. I can't tell what some of these are - e.g. 'GridCellTorus' or 'disentangle.' I can't find what these paths correspond to.

- I had issues with installing the packages listed in env.yaml (cudnn==8.8 had to be set to just cudnn, and hickle= ... needed to be hickle==). Eventually the code errored out at trying to build the wheel for mpi4py.

- A note regarding the data folder: in the readme, it seems that there is some confusion about what the variables in the uploaded data correspond to (e.g. the spike times, the variable z, etc). It might be worth verifying with the authors of the dataset any outstanding questions, just so that the authors can be confident that they are using the variables correctly.

Minor comments

- A question regarding GPKR - does the user have to set the number of label variables? In other words, does the user have to already know the intrinsic dimensionality of the neural population?

- The authors state that they create speed-balanced datasets prior to doing any analyses. I don't think this is an issue, but it would be nice if they could be clearer about what sort of issues in their analyses might have arisen if they didn't do this processing on the data. Were they concerned that speed-imbalanced data would generate a less interpretable manifold, especially if they didn't know a priori how speed would influence the shape of the manifold?

- There were quite a few typos and sentences that were not grammatically correct throughout the text. This might get corrected later in the review process, but I think it's worth doing a thorough read-through to eliminate these errors, since it will have to happen at some point.

Reviewer #1 (Remarks on code availability):

I did not go through and review all the code extensively, but rather I tried to run the GKR example. I wasn't able to get the code running unfortunately. I detailed the issues I had in the review above. The

code does include a readme, although I think the readme could be edited to be more helpful.

Reviewer #2 (Remarks to the Author):

In this manuscript, Ye and Wessel address the important issue of how a rodent's velocity affects information encoding in the medial entorhinal cortex's grid cells, and also exposes whether the "noise" dimensions are orthogonal or not.

The manuscript sells itself as a methods paper, touting the "Gaussian Kernel Regression" as novel. While I am not an expert on Gaussian process modeling, I think this claim is a stretch. From what I understand, the first step is to let the mean of the GP be zero, and then to fit the covariance matrix. This is a perfectly reasonable approach, of course. I suspect the details might differ, but this approach does bear some similarities to Rule et al and Papastathoopoulos et al, which build upon other papers' approaches.

It might be useful to draw a more detailed comparison to these other papers in the Methods section.

The presentation in the manuscript is clear and detailed, but at times I had the impression that the authors wanted to throw the kitchen sink at the problem. For instance, what is the point of doing topological data analysis on the homology classes of the filtrations to get the "bar codes", but then to stop short of using cohomology to decode? If velocity is a new variable in addition to the spatial coordinates, then what is the topology of this space as a whole? The authors refer to manifold dilation, which would invoke concepts from differential geometry, but topology is only concerned with continuity (And I do have to object to the claim that a Betti sequence of (1,2,1) indicates a torus--it can be a torus, but Betti sequences from the abelianization of algebraic geometry do not map one-to-one to unique topologies)

My recommendation: rewrite the main text to focus solely on the questions of neuroscientific import (and leave the methods section as is) and perhaps drop the topological data analysis to simplify the manuscript.

The main result, namely that the spatial information increases with speed, has sufficient merit to carry the entire manuscript. I do not think the authors should stress the "methods paper" aspect.

Reviewer #3 (Remarks to the Author):

This paper is well-written and investigates how running speed modulates the information coding of grid cells. Using experimental data from previous study in Moser's Lab, the authors used a novel method, GKR, to infer the manifold and noise covariance of neural data. They showed that increasing speed will increase spatial accuracy as well as noise covariance (that is shown to be detrimental). However, the overall effect is the improvement of spatial coding, as shown in the increase of Fisher Information. GKR, potentially coupled with dimension reduction techniques, will provide a useful tool to understand neural population coding.

Some comments are:

* What is the implication of the results on the underlying model of grid cells?

* Figure 1C: is the increase of speed accuracy caused by the more divided spatial positions when the speed is large?

- * Figure 1: Will the choice of decoder (other than a logistic regressor) affect the result?
- * Equation (1): Does the assumption of GKR always hold (i.e., the data always follows a Gaussian distribution)?
- * Figure 4C: is there any reason why S1M1 has negative slope?
- * I'm curious about how this paper would reveal the coding properties of hippocampal cells and how it's different from grid cell investigated here? (If authors can test it straightaway, that would be awesome; but it doesn't imply authors have to do it in this paper)
- * Line 346: "do not"

Reviewer #4 (Remarks to the Author):

This study investigates how running speed affects spatial coding in grid cells. The researchers developed a novel method called Gaussian Process with Kernel Regression (GKR) to analyze simultaneously recorded grid cell population activity from an information geometry perspective. Their key findings show that:

1. Higher running speeds lead to better spatial coding accuracy
2. Running speed increases both the size of the grid cell manifold (geometric representation) and noise levels
3. The positive effect of manifold expansion outweighs the negative effect of increased noise
4. Cell-to-cell noise correlations are detrimental to information coding

Strengths:

1. Novel methodology (GKR) that allows analysis of high-dimensional neural data without requiring repeated trials
2. Comprehensive validation of the method using synthetic data
3. Rigorous mathematical framework and statistical analysis
4. Clear connection between theoretical predictions and experimental results

I have two main concerns:

1. Choice of labels:

- The method assumes we know the relevant variables (position, velocity) a priori
- It imposes a particular structure on how information is encoded
- Could miss other important variables or combinations of variables
- Assumes separability of these variables

Specific impacts on analysis:

- Could artificially separate variables that are actually encoded jointly
- Might miss nonlinear combinations of variables
- Risk of confirmation bias - finding structure we assume exists
- Could underestimate the dimensionality of the true representation

Possible mitigations:

- Compare different label choices using model selection
- Use nonparametric approaches to discover relevant variables
- Test for interactions between different labels

- Validate findings using behavioral relevance

2. Choice of noise model:

- Real neural noise may be non-Gaussian
- Could have heavy tails or be skewed
- May vary with the signal
- Could have complex temporal structure

Impact on analysis:

- Could misestimate uncertainty
- May not capture true noise correlations
- Risk of over/under-fitting
- Could affect estimates of Fisher information

Possible mitigations:

- Compare different noise models (e.g., Poisson, Log-normal)
- Use nonparametric bootstrapping approaches
- Test robustness to noise model assumptions
- Validate predictions using held-out data

Other weaknesses I would like to see discussed:

1. The study focuses primarily on mathematical/theoretical aspects, with limited discussion of biological implications

a. Are there experiments where manipulating running speed artificially (e.g., treadmill) could establish causality, e.g. Campbell 2021

b. How can the method generalize to other brain regions or behaviors when one has to assume labels

c. Grid cells are known to react to environmental change (e.g. Barry et al 2012) How can environmental complexity or novelty interact with speed effects

2. Limited exploration of the mechanisms behind speed-dependent improvements in spatial coding

A. Expand the discussion of biological mechanisms that might explain the speed-dependent effects

3. Include more detailed discussion of the limitations of the GKR method and potential failure cases

Minor:

In the first sentence you write grid cells were recorded, but this study only uses data from a previous study, please modify this sentence to make it immediately clear that you used data from this reference.

Responses to Reviewer Comments

We thank the Reviewers for their constructive feedback. Please find below our point-by-point responses to the Reviewers' comments. We have revised the manuscript accordingly and highlighted those changes in green.

Reviewer #1, summary of main findings

In this work, the authors examine the population level activity of grid cells through a new method they develop here, Gaussian process with kernel regression (GPKR). GPKR infers both the manifold (low-dimensional structure) of the data as well as the noise covariance. Through GPKR, they examine the influence of running speed the grid cell code, including how running speed correlates with the noisiness of firing. They analyze a previously published dataset of simultaneously recorded grid cells from Gardner et al., 2022. Consistent with prior analyses on this dataset, they find that GPKR reveals a torus manifold. They find that running speed dilates this manifold, which increases the fidelity of the spatial code, and that running speed increases the noisiness of firing, which decreases encoding accuracy. However, when looking at the two effects combined, the encoding accuracy net increases with running speed. In addition, they use GPKR to examine the impact of noise correlations on the spatial code. They find that the noise correlations are generally detrimental, meaning that the noise lies along the manifold and thus could be 'confused' for real signal by a decoder.

Overall, the work presented was interesting and easy to follow. In terms of impact, the authors present both an interesting manifold-discovery technique suited for single trial data and scientific results on grid code coding. Although I am not an expert on the development of techniques like GPKR, what they present seems reasonable and, from a very high level, relatively clear (that they iteratively optimize the mean and the covariance structure). It is also great that they included a GitHub link so that the code/analyses can be explored and used by others. However, on the scientific results side, it is unclear to me how novel or surprising their presented results are. More detailed comments on that, and on a few other items, are detailed below.

We thank the reviewer for their interests in our work.

Reviewer #1, specific comments

***RI.1.** Grid cell firing rates tend to increase with running speed. The authors also mention this in the discussion. It is not clear to me how much this effect explains the results they see, or how they interpret their results considering this fact. If we also assume that neurons are Poisson-like in that the variance scales with the mean, would these two facts completely explain both the dilation of the manifold and the increase in noise with running speed? In other words, if the authors were to simulate grid cells according to Poisson firing and assuming that firing rate increases with running speed, would they see the same effects? And if so, would the authors interpret their findings to visualize *how* running speed increases coding fidelity? Or, perhaps as another way to visualize or quantify the change in the noise with running speed? As I write this, I am wondering if the authors view their own paper as more of a technique paper, or a scientific results paper, or somewhere in between.*

This is a very good question. We investigate whether some key findings—namely, the speed-dependent modulations of manifold size, noise, and Fisher information (Figures 3E, 4C, 5B)—can be explained by an idealized Poisson grid cell model with positive speed gain. Specifically, we assume an independent Poisson Speed-Gain (IPSG) grid cell population where each grid cell’s response $r(\mathbf{x}, v)$ is:

$$r(\mathbf{x}, v) \sim \text{Poisson}(f(v)M(\mathbf{x})), \quad (1)$$

where \mathbf{x} represents position, v denotes speed, and $f(v)M(\mathbf{x})$ defines the mean firing rate, with $M(\mathbf{x})$ indicating the spatial rate map, and $f(v)$ is a monotonically increasing function of speed that modulates the entire rate map.

We then conducted both analytical and numerical analyses of IPSG grid cells (Methods: *Independent Poisson Speed-Gain (IPSG) grid cells; Simulation of IPSG grid cells*). Consistent with the reviewer’s conjecture, IPSG grid cells exhibit similar speed modulation effects (SI Figure 12) agree with some major findings in our manuscript (Figure 3E, 4C, 5B).

However, while the IPSG model qualitatively accounts for speed modulation effects, it is a highly simplified model with unreliable assumptions about the real grid cells. For instance: (1) Speed modulation may not be a simple gain factor. (2) The noise statistics of grid cells may not follow a Poisson distribution¹. (3) Grid cell firings are correlated². Ignoring noise correlation will lead to very different estimation of projected noise and Fisher information values (as suggested by the large value differences between GKR and IFCG’s GKR, Figure 6). Therefore, it is unknown *a priori* what the speed modulation effects on the population data would be. Further, IPSG cannot be used to study the function of noise correlation on coding (Figure 6).

Thus, we agree with the reviewer that our findings share certain consistency with previous studies on individual grid cells, but due to the complexity of the neural system, we think it is necessary and important to perform direct analysis on the population data. Therefore, we consider our manuscript as partially method paper and partially scientific results paper.

We appreciate the reviewer’s insightful question, which connects population-level analysis with individual neurons³⁻⁵. We have incorporated the relevant results about IPSG in Ln.362–376, SI Figure 12, with Methods details described in *Methods: Independent Poisson Speed-Gain (IPSG) grid cells; Simulation of IPSG grid cells* at page 29–30.

R1.2. *Along somewhat similar lines, regarding the other major scientific result presented - that the noise correlations in grid cells lie along the manifold and thus are detrimental to the fidelity of the spatial code - are surprising given what we know about grid cells. In this case, much prior work has indicated that grid cell activity arises from connectivity between cells, and thus noise is most likely correlated between cells (and not independent). I’m not sure the fact that this is a strong prediction (that noise is correlated across neurons and detrimental) is an issue, but it’s just a bit unclear in this paper if the authors view this as a novel finding or an illustration of their technique.*

We agree with the reviewer that it is natural that grid cell firing are correlated, as has been shown previously by (for example) Nagele et al. 2020¹

However, the impact of noise correlations on neuronal population coding is unclear. Recent work indicates that noise correlations can be detrimental or beneficial, depending on their structure and the geometry of the underlying manifold⁶⁻⁸. The study of noise structure is a vibrant research area⁹.

To illustrate this duality of noise correlation effects, we simulated a simple two-neuron system (SI Figure 13). The simulation shows that, depending on the geometry, noise correlations can either degrade or enhance neural coding.

With respect to this, the contribution of our manuscript in Figure 6 is to perform a direct geometric analysis on the data and to show that the effects of noise correlation is “shaping” the noise covariance such that more noise is projected onto the manifold surface (Figure 6C), thus leading to detrimental coding (Figures 6D, 6E). We think these are novel findings.

To clarify our motivation and results, we have added a simple two-neuron system for illustration (Methods: *A two-neuron example demonstrating the effects of noise correlation* at Ln.1008–1032; SI Figure 13; and main text Ln. 385).

RI.3. This paper analyzes a previously published dataset, but the authors do not make this entirely clear from the outset - the language in the main text is vague as to where the data came from (although it indicates it's from Gardner in the methods). I would suggest changing the wording so it's clear what is being analyzed.

We thank the reviewer highlighting this. We have revised the sentence into (also see Ln89):

We analyzed the grid cell activities of rats previously recorded by Gardner et al. (2022) during open-field (OF) foraging tasks.

RI.4. I went through the code on GitHub. I didn't run attempt to the code on the Gardner data, which I see must be downloaded from their site directly. I did, however, try to run the GKR example. Unfortunately, I wasn't able to install the environment as provided in env.yaml. I detailed my notes below - I think there are several places where the implementation can be more user friendly. It's possible that the failure to download was on me (I didn't have a ton of time to devote to troubleshooting), but it might be worth having another novel user try to download and run the code. I did, however, try to run the GKR example. Unfortunately, I wasn't able to install the environment as provided in env.yaml. I detailed my notes below - I think there are several places where the implementation can be more user friendly.

We thank the reviewer for the suggestions to improve the readability of our code. In response, we have revised the *README.md* and simplified the code usage (see speed_grid_cell_information). Here are a few points we want to highlight.

For users who are only interested in using GKR, we have prepared an standalone file, *GKR_demo.ipynb*, which contains all the necessary code to run GKR, along with a simple toy example. Users can run *GKR_demo.ipynb* on Colab, or execute it locally after installing a minimal set of dependencies via: `pip install -r requirements_gkr_demo.txt`.

For reproducing the main figures, users need to install the environment by `pip install -r requirements.txt`, then run the codes in *paper.ipynb*.

Below, we provide more detailed point-by-point responses.

but it might be worth having another novel user try to download and run the code.

Thanks for the suggestion. We have asked a novel user to test our code, Dr. Haoran Li, who has also been added to the acknowledgments.

It seems that the code as uploaded was developed on Windows and might be only compatible with Windows OS (e.g. the `add_python_path`). If true, it would be helpful if the authors stated that on the github main page.

The code can be run on both Windows and Linux. We have revised the code structure and *README.md* to make the workflow independent of the operating system.

Also, it looks like this code needs Nvidia GPUs – this and any other hardware dependencies would be nice to state upfront.

The code can run on both CPUs and GPUs, with GPUs as the default option if they are available.

To run code on GPUs, besides installing *requirements.txt*, the users also need to install CUDA, as indicated by the new *README.md*.

in the code ‘`add_python_path`’, there are a lot of hardcoded paths for the authors’ directories. I can’t tell what some of these are – e.g. ‘GridCellTorus’ or ‘disentangle.’ I can’t find what these paths correspond to.

We apologize for the confusion. The code in ‘disentangle’ was used for code experimentation but is not required for running any code in the published version of the GitHub repository. To avoid ambiguity, we have completely removed *add_python_path*.

There’s no need to set Python path in this new version.

I had issues with installing the packages listed in `env.yaml` (`cuda==8.8` had to be set to just `cuda`, and `hickle= ...` needed to be `hickle==`). Eventually the code errored out at trying to build the wheel for `mpi4py`.

We appreciate the reviewer's bug reporting. We have removed *env.yaml* and switched to using pip (*pip install -r requirements.txt*). We have tested *requirements.txt* on a new machine.

A note regarding the data folder: in the readme, it seems that there is some confusion about what the variables in the uploaded data correspond to (e.g. the spike times, the variable `z`, etc). It might be worth verifying with the authors of the dataset any outstanding questions, just so that the authors can be confident that they are using the variables correctly.

We appreciate the reviewers' suggestions. We have contacted Dr. Rich Gardner (the first author of Gardner et al. 2022) to verify information, and confirmed that the data variables were understood correctly. We have added Dr. Rich Gardner into the acknowledgement (page 34).

RI.5. *A question regarding GPKR - does the user have to set the number of label variables? In other words, does the user have to already know the intrinsic dimensionality of the neural population?*

This is a good question. While users need to know the label variables of interest, they do not need to know all labels; therefore, they don't need to know the intrinsic dimensionality.

Intuitively, one could consider that the data is modeled by multiple latent variables (a.k.a. labels), but the user is only interested in the representational properties of a specific subset of labels (which should be known). GKR enables the inference of the “sub-manifold” that encodes these label variables of interest, while treating the influence of other latent variables (which can be unknown) as effective “noise” in relation to the label-of-interest.

For example, in our study, we are interested in the representation of speed and location. We applied GKR to infer the geometric structure relevant to these variables, while the effects of other latent factors, such as mental states, were incorporated into the noise term.

To clarify this, we have explicitly mentioned the usage of GKR in Ln.158–163.

R1.6. The authors state that they create speed-balanced datasets prior to doing any analyses. I don't think this is an issue, but it would be nice if they could be clearer about what sort of issues in their analyses might have arisen if they didn't do this processing on the data. Were they concerned that speed-imbalanced data would generate a less interpretable manifold, especially if they didn't know a priori how speed would influence the shape of the manifold?

We balanced the data because of the principle of a controlled experiment, rather than the GKR method. The objective of this study is to compare neural representations at different speeds, which requires controlling nuisance factors such as the number of data points.

Failing to balance the data could lead to misleading conclusions. For instance, we used Spatial Coding Accuracy (SCA) to assess spatial coding quality (Figures 1B, 1C). SCA is based on logistic regression, whose performance depends on the number of training data points. Without balancing, SCA values would be artificially higher in speed bins with more data points.

We appreciate the reviewer's comment on this point and have revised the main text (Ln.97–99) to clarify our rationale for using speed-balanced data.

R1.7. There were quite a few typos and sentences that were not grammatically correct throughout the text. This might get corrected later in the review process, but I think it's worth doing a thorough read-through to eliminate these errors, since it will have to happen at some point.

We thank the reviewer's suggestion. We have thoroughly reviewed the manuscript and corrected grammatical errors as seen in the newly uploaded version.

R1.12. I did not go through and review all the code extensively, but rather I tried to run the GKR example. I wasn't able to get the code running unfortunately. I detailed the issues I had in the review above. The code does include a readme, although I think the readme could be edited to be more helpful.

We thank the reviewer highlighting issues in code. Please see responses above (R1.4) for our revisions on the code and readme.

Reviewer #2, summary of main findings

In this manuscript, Ye and Wessel address the important issue of how a rodent's velocity affects information encoding in the medial entorhinal cortex's grid cells, and also exposes whether the "noise" dimensions are orthogonal or not.

We thank the reviewer for their interest in our work.

Reviewer #2, specific comments

R2.1. *The manuscript sells itself as a methods paper, touting the "Gaussian Kernel Regression" as novel. While I am not an expert on Gaussian process modeling, I think this claim is a stretch. From what I understand, the first step is to let the mean of the GP be zero, and then to fit the covariance matrix. This is a perfectly reasonable approach, of course. I suspect the details might differ, but this approach does bear some similarities to Rule et al and Papastathopoulos et al, which build upon other papers' approaches.*

It might be useful to draw a more detailed comparison to these other papers in the Methods section.

We thank the reviewer for highlighting these interesting papers. But after careful reading, we propose that their methods are different from ours, as reasons listed below.

First, the objectives of the methods are very different. Our method aims to infer cell-to-cell properties, such as neural population noise covariance. In contrast, both Rule et al.¹ and Papastathopoulos et al.² focus on single-cell statistics and do not infer inter-cell properties. Notably, the term “covariance” in their context refers to single-cell firing variability across different locations, whereas in our work, “covariance” denotes cell-to-cell noise covariance.

Second, the mathematical formulations differ. Our method has two kernels: one for fitting the Gaussian process to the neural population data and another applied to the residual vector to infer cell-to-cell noise covariance. In contrast, the methods of Rule et al. and Papastathopoulos et al. use a single kernel for the Gaussian process to infer single neural statistics, while the single-grid-cell noise is determined by the Gaussian process output. Such modeling does not take into account cell-to-cell noise correlation—which is perfectly fair in their papers since studying neural population is not their purpose.

We agree with the reviewer that it would be useful to compare these interesting methods in our manuscript, please refer a simplified version of the above discussion in Ln.711–713.

R2.2. *The presentation in the manuscript is clear and detailed, but at times I had the impression that the authors wanted to throw the kitchen sink at the problem. For instance, what is the point of doing topological data analysis on the homology classes of the filtrations to get the "bar codes", but then to stop short of using cohomology to decode? If velocity is a new variable in addition to the spatial coordinates, then what is the topology of this space as a whole? The authors refer to manifold dilation, which would invoke concepts from differential geometry, but topology is only concerned with continuity (And I do have to object to the claim that a Betti*

sequence of (1,2,1) indicates a torus---it can be a torus, but Betti sequences from the abelianization of algebraic geometry do not map one-to-one to unique topologies)

My recommendation: rewrite the main text to focus solely on the questions of neuroscientific import (and leave the methods section as is) and perhaps drop the topological data analysis to simplify the manuscript.

We thank the reviewer's suggestions. Please see below for point-to-point response:

what is the point of doing topological data analysis on the homology classes of the filtrations to get the "bar codes", but then to stop short of using cohomology to decode?

Before explaining our motivation, we would like to mention that we agree with the reviewer that topological analysis is not essential to our paper's conclusion. We have significantly simplified the texts on topological analysis to avoid unnecessary confusion: (1) See the simplified text at Ln.225–235; (2) We removed the bar code panel in Figure 3 but still showed it in SI Figure 8.

We have two main motivations about our topological analysis. First, previous work (Gardner et al. 2022) has demonstrated that using data clouds, the topology of grid cell responses exhibits a barcode (1, 2, 1). We were curious whether the manifold inferred by our method, GKR, could reproduce a similar barcode. Indeed, our results showed similar barcodes (SI Figure 8).

Second, some readers may not be familiar with the prior work by Gardner et al. (2022). We aim to include topological analysis to provide some intuitions.

We have stopped using cohomology for decoding because, as the reviewer also suggested, this decoding is not very relevant to the main results (speed modulation of population representation) in our manuscript. Moreover, we have already performed decoding using logistic regression earlier in our manuscript (Figure 1).

If velocity is a new variable in addition to the spatial coordinates, then what is the topology of this space as a whole?

This is a good question. We first used GKR to infer the manifold, which consists of three intrinsic coordinates: x-position, y-position, and speed. Next, we sampled a random mesh of points from the inferred manifold. Persistent homology indicates that the barcodes are mostly (1, 2, 1) (SI Figure 8). This result is consistent with Gardner et al 2022, as they also did not restrict their data to a fixed speed.

The authors refer to manifold dilation, which would invoke concepts from differential geometry, but topology is only concerned with continuity (And I do have to object to the claim that a Betti sequence of (1,2,1) indicates a torus---it can be a torus, but Betti sequences from the abelianization of algebraic geometry do not map one-to-one to unique topologies)

We agree with the reviewer that topological analysis concerns continuity that is not much relevant to the focus of our manuscript. To make our writing more concise, we decided to follow the reviewer's suggestion and simplified the text on topological analysis (Ln.225–235).

R2.3. My recommendation: rewrite the main text to focus solely on the questions of neuroscientific import (and leave the methods section as is) and perhaps drop the topological data analysis to simplify the manuscript.

The main result, namely that the spatial information increases with speed, has sufficient merit to carry the entire manuscript. I do not think the authors should stress the "methods paper" aspect.

We thank the reviewer for their suggestion. We have simplified that part of the topological analysis (Figure 3, Ln.225–235, while leaving the methods section as is) to make the manuscript more straightforward. We also added more biological interpretation in the discussion to slightly shift the emphasis from a method paper to a neuroscientific paper (Ln.473–495).

However, after careful consideration, we think our method has its own novelty compared to earlier methods, as explained in R2.1, and is practically useful (also see the shared code https://github.com/AgeYY/speed_grid_cell_information.git). We would like to still keep our manuscript as a partially neuroscientific and partially method paper.

Reviewer #3, summary of main findings

This paper is well-written and investigates how running speed modulates the information coding of grid cells. Using experimental data from previous study in Moser's Lab, the authors used a novel method, GKR, to infer the manifold and noise covariance of neural data. They showed that increasing speed will increase spatial accuracy as well as noise covariance (that is shown to be detrimental). However, the overall effect is the improvement of spatial coding, as shown in the increase of Fisher Information. GKR, potentially coupled with dimension reduction techniques, will provide a useful tool to understand neural population coding.

We thank the reviewer for their interests in our work.

Reviewer #3, specific comments

R3.1. What is the implication of the results on the underlying model of grid cells?

This is a good question. There are two major models of grid cells¹⁰: the rate-based model and the oscillatory-interference model. In the rate-based model, running speed is an input to grid cell network⁵. Higher running speeds may increase firing rates^{3,4}, which may sharpen the rate map, reduce the signal to noise ratio and hence improve the overall population Fisher information (Figure 5). Alternatively, in the oscillatory-interference model, running speed modulates the frequency of membrane potential which might lead to more accurate spatial fields^{10,11}. Finally, higher running speeds imply more frequent encounters with environmental boundaries, allowing for more frequent corrections in grid coding¹². Despite these conjectures, it should be noted that real grid cells are complicated, involving correlated noise. Models based solely on individual grid cells, without accounting for noise correlations, may result in substantial estimation errors, as shown by the pronounced discrepancies between the original data and the IFGC model in Figure 6. The connection between population results and individual grid cells remains for future exploration.

We have added the above discussion to Ln.460–472.

R3.2. Figure 1C: is the increase of speed accuracy caused by the more divided spatial positions when the speed is large?

The spatial separation between the two boxes (Figure 1B) was fixed at 10 cm for all speed values. We have updated the main text (Ln.128) and the Figure 1 caption (Ln.106) to clarify this. Please also refer more details in the Methods section (Ln.602).

R3.3. Figure 1: Will the choice of decoder (other than a logistic regressor) affect the result?

To explore this, we employed a support vector machine and a perceptron to compute the SCA. The results remain robust (SI Figure 3).

We have updated the text (Ln 144–145) and revised SI Figure 3 to incorporate this new result.

R3.4. Equation (1): Does the assumption of GKR always hold (i.e., the data always follows a Gaussian distribution)?

This is a good question. We performed a normality test on the data and found that it does not follow a Gaussian distribution (see text in the Results section, Ln.178; *Methods section: Testing the normality assumption in data* at Ln766–781; and SI Figure 4). However, this does not invalidate the results of GKR, as the Gaussian distribution is a commonly used approximation¹³.

To assess the robustness of our results, we designed an alternative method—Gaussian process regression with kernel sampling (GKR-S)—which does not assume data normality. GKR-S is similar to GKR in that it uses a Gaussian process to infer a smooth mean, but it estimates the noise distribution through resampling without assuming normality (see *Methods: Gaussian process regression with kernel sampling (GKR-S)* at Ln.782–818). GKR-S performs well when the data is low-dimensional (see SI Figure 6A). Therefore, to apply GKR-S to the grid cell dataset, we first performed dimensionality reduction. We then applied GKR-S to this dataset using the same procedure as in the main manuscript.

We found that GKR-S, which is capable of capturing non-normal structures (SI Figure 6B), closely agrees with the results of GKR. Therefore, the findings in our manuscript remain robust.

R3.5. Figure 4C: is there any reason why S1M1 has negative slope?

We found that the negative slope result is robust. Even after preprocessing the dataset with dimensionality reduction and using GKR-S instead of GKR, the mean slopes remain negative (SI Figures 6E and 6F).

What is the underlying reason? We are not sure. The number of grid cells in S1M1 ($n = 96$) is within a reasonable range ($n=50$ to 160 across all datasets we inspected); and the rate maps of S1M1 grid cells do not appear particularly distinctive (SI Figure 1). Therefore, we conjecture that this phenomenon may be due to some intrinsic random factors in the formation of the S1M1 module or to the relatively small number of recorded grid cells compared to the full grid cell population in the brain. The negative slope remains an open question.

R3.6. I'm curious about how this paper would reveal the coding properties of hippocampal cells and how it's different from grid cell investigated here? (If authors can test it straightaway, that would be awesome; but it doesn't imply authors have to do it in this paper)

This is an interesting question. Please see our preliminary thoughts below.

In theory, grid cells are modeled as a primary feedforward input to the hippocampus^{14,15}, suggesting that running speed should also enhance place cell representation as well. However, this feedforward model is a simplification, as the hippocampus sends feedback projections to the MEC¹⁶. Moreover, place cells receive inputs not only from grid cells but also from other sources, such as head direction cells¹⁴. These additional mechanistic factors obscure how running speed modulates place cell activity. Indeed, earlier research suggests that the majority of place cells are not strongly modulated by running speed¹⁷. Nevertheless, beyond speed modulation, movement direction could influence the representational geometry of place cells, as it has been shown to reshape place fields¹⁸. Extending geometric analyses to other cell types remains an interesting avenue.

We have a similar discussion into Ln.486–496.

R3.7. Line 346: "do not"

We thank the reviewer's correction. We have corrected "do no" to "do not", see Ln340.

Reviewer #4, summary of main findings

This study investigates how running speed affects spatial coding in grid cells. The researchers developed a novel method called Gaussian Process with Kernel Regression (GKR) to analyze simultaneously recorded grid cell population activity from an information geometry perspective. Their key findings show that:

- 1. Higher running speeds lead to better spatial coding accuracy*
- 2. Running speed increases both the size of the grid cell manifold (geometric representation) and noise levels*
- 3. The positive effect of manifold expansion outweighs the negative effect of increased noise*
- 4. Cell-to-cell noise correlations are detrimental to information coding*

Strengths:

- 1. Novel methodology (GKR) that allows analysis of high-dimensional neural data without requiring repeated trials*
- 2. Comprehensive validation of the method using synthetic data*
- 3. Rigorous mathematical framework and statistical analysis*
- 4. Clear connection between theoretical predictions and experimental results*

We thank the reviewer for their interests in our manuscript.

Reviewer #4, specific comments

R4.1. Choice of labels:

- (a) The method assumes we know the relevant variables (position, velocity) a priori*
- (b) It imposes a particular structure on how information is encoded*
- (c) Could miss other important variables or combinations of variables*
- (d) Assumes separability of these variables*

Specific impacts on analysis:

- (e) Could artificially separate variables that are actually encoded jointly*
- (f) Might miss nonlinear combinations of variables*
- (g) Risk of confirmation bias - finding structure we assume exists*
- (h) Could underestimate the dimensionality of the true representation*

Possible mitigations:

- (i) *Compare different label choices using model selection*
- (j) *Use nonparametric approaches to discover relevant variables*
- (k) *Test for interactions between different labels*
- (l) *Validate findings using behavioral relevance*

These are good questions. We categorized these questions into two major questions: (1) (*relevant to question a, b, c, f, h, i, j*) our method requires explicit label, which can miss other latent labels (i.e., variables); (2) (*relevant to question d, e, g, k*) the labels (position and speed) might be correlated, hence may introduce unexpected manifold structure.

(1) *our method requires explicit label, which can miss other latent labels (i.e., variables)*

We would like to clarify that the goal of GKR is not to uncover all structures within the data, but rather to identify the substructure (i.e., “submanifold”) relevant to the label of interest. For example, in our case, we focus on x-position, y-position, and speed. We use GKR to infer the corresponding submanifold, while variations due to other latent variables are treated as equivalent “noise” in the encoding of x, y, and speed. This approach allows us to further estimate, for instance, the Fisher information on x and y. A side advantage of focusing on the submanifold is that it may be simpler than the full manifold, making it easier to interpret.

Despite of this, we agree the reviewer’s that GKR cannot infer latent labels. This is a limitation of our method for people who are interested in latent structure. We have added our discussion to indicate this limitation in Line 446–459.

Together, we added Ln.97–99 to clarify that the primary purpose of GKR is to study the representation properties of the label of interest, and included a discussion of GKR’s limitations in Ln.446–459.

(2) *The labels (position and speed) might be correlated, hence may introduce unexpected manifold structure.*

We computed the behavioral correlation between position and speed and found that all correlations were very small (median absolute correlation = 0.085, SI Figure 7). This suggests that our results are unlikely to due to the correlations between different labels.

We have added SI Figure 7 and revised Ln. 225 to indicate this.

R4.2. Choice of noise model:

- *Real neural noise may be non-Gaussian*
- *Could have heavy tails or be skewed*
- *May vary with the signal*
- *Could have complex temporal structure*

Impact on analysis:

- *Could misestimate uncertainty*
- *May not capture true noise correlations*
- *Risk of over/under-fitting*
- *Could affect estimates of Fisher information*

Possible mitigations:

- *Compare different noise models (e.g., Poisson, Log-normal)*
- *Use nonparametric bootstrapping approaches*
- *Test robustness to noise model assumptions*
- *Validate predictions using held-out data*

This is a good question. We also thank the reviewer's suggestions.

We performed a normality test on the data and found that it does not follow a Gaussian distribution (see text in the Results section, Ln.178; *Methods section: Testing the normality assumption in data* at Ln766–781; and SI Figure 4). However, this does not invalidate the results of GKR, as the Gaussian distribution is a commonly used approximation¹³.

To assess the robustness of our results, we designed an alternative method—Gaussian process regression with kernel sampling (GKR-S)—which does not assume data normality. GKR-S is similar to GKR in that it uses a Gaussian process to infer a smooth mean, but it estimates the noise distribution through resampling without assuming normality (see Methods: *Gaussian process regression with kernel sampling (GKR-S)* at Ln.782–818). GKR-S performs well when the data is low-dimensional (see SI Figure 6A). Therefore, to apply GKR-S to the grid cell dataset, we first performed dimensionality reduction. We then applied GKR-S to this dataset using the same procedure as in the main manuscript.

We found that GKR-S, which is capable of capturing non-normal structures (SI Figure 6B), closely agrees with the results of GKR. Therefore, the findings in our manuscript remain robust.

R4.3. *The study focuses primarily on mathematical/theoretical aspects, with limited discussion of biological implications*

R4.3a. *Are there experiments where manipulating running speed artificially (e.g., treadmill) could establish causality, e.g. Campbell 2021*

It would be ideal to have a speed-controlled experiment (e.g., a treadmill¹⁹ or a bottomless car²⁰) combined with large-scale grid cell recordings. However, to our knowledge, existing speed-controlled experiments primarily involve recordings from MEC cells in general^{19,20}. The number of single-module grid cells in such experiments may be insufficient to reliably reveal meaningful geometric structures.

R4.3b. How can the method generalize to other brain regions or behaviors when one has to assume labels

This is an interesting question; we think there are two major ways to apply GKR.

First, GKR can be applied in cases where the labels are known. Such scenarios usually occur in vision research. For example, visual stimuli usually constructed with explicitly known parameters (e.g. orientation of a grating stimuli), these parameters can serve as labels.

Second, GKR can be applied in cases where the labels are unknown but can be inferred through other methods. Several methods can infer latent variables, ranging from simple approaches like principal component analysis to nonlinear methods such as Spline Parameterization for Unsupervised Decoding (SPUD)²¹ and deep neural networks like variational autoencoder²².

Overall, while we agree with the reviewer that GKR's requires label is a limitation, we believe it remains quite useful in many scenarios. We have incorporated this point into a discussion paragraph in Ln.452–459.

*R4.3c. Grid cells are known to react to environmental change (e.g. Barry et al 2012)
How can environmental complexity or novelty interact with speed effects*

This is an interesting question. Rats typically exhibit higher running speeds in novel environments²³, as implied by this paper, which might enhance grid coding thus supporting more effective adaptation to novel surroundings. Grid cell representations are known to change in novel environments^{24–26}. Some studies suggest that the grid pattern rescales in a novel environment (e.g., Barry et al. 2012²⁷); some others propose that rats refine their grid coding by learning the environment's boundaries^{12,26}. These alterations in individual grid cell patterns may correspond to changes in the representation geometry, such as a rescaling of the toroidal structure or localized distortions near environmental boundaries. The effects of environmental modulation on population-level representations remain an open question for future investigation.

We have added the discussion into the manuscript's Discussion section (Ln.473–480).

R4.4. Limited exploration of the mechanisms behind speed-dependent improvements in spatial coding

A. Expand the discussion of biological mechanisms that might explain the speed-dependent effects

This is a good question. There are two major models of grid cells¹⁰: the rate-based model and the oscillatory-interference model. In the rate-based model, running speed is an input to grid cell network⁵. Higher running speeds may increase firing rates^{3,4}, which may sharpen the rate map, reduce the signal to noise ratio and hence improve the overall population Fisher information (Figure 5). Alternatively, in the oscillatory-interference model, running speed modulates the frequency of membrane potential which might lead to more accurate spatial fields^{10,11}. Finally, higher running speeds imply more frequent encounters with environmental boundaries, allowing for more frequent corrections in grid coding¹². Despite these conjectures, it should be noted that real grid cells are complicated, involving correlated noise. Models based solely on individual

grid cells, without accounting for noise correlations, may result in substantial estimation errors, as shown by the pronounced discrepancies between the original data and the IFGC model in Figure 6. The connection between population results and individual grid cells remains for future exploration.

We have added the above discussion to Ln.460–472.

R4.5. Include more detailed discussion of the limitations of the GKR method and potential failure cases

Thanks for the suggestions. We have included the following discussion in Ln.446–459.

However, GKR has its limitations. First, it does not perform well in very high-dimensional spaces, which may require a larger number of data points (Figure 2 and SI Figure 5). This issue may be mitigated by first applying a dimensionality reduction method to the data²⁸. Second, GKR assumes that the data follows a normal distribution. While the normal distribution is a commonly used approximation^{13,29}—GKR may produce unreliable results if the true distribution deviates significantly from normality. It is advisable to perform a normality test (SI Figure 4), compute test data's likelihood³⁴, or use an alternative method (e.g., GKR-S, SI Figure 6) to verify the results. Finally, GKR is only applicable when the data explicitly contains labels. In other words, its purpose is to evaluate the geometric representation properties of known labels of interest. For example, in vision³⁰, navigation, or working memory³¹ studies, the labels of interest are often defined by stimulus parameters. However, in cases where one aims to evaluate the representation structure of latent variables, it is necessary to first apply latent variable inference methods^{22,28}, then using GKR. Overall, future improvements to GKR could focus on enhancing its performance in high-dimensional settings, adapting it for non-normal data, and extending its applicability to scenarios without explicit labels.

R4.6. In the first sentence you write grid cells were recorded, but this study only uses data from a previous study, please modify this sentence to make it immediately clear that you used data from this reference.

Thanks for reminding. We have corrected it (Ln.89) as:

We analyzed the grid cell activity of rats recorded by Gardner et al. (2022) during open-field (OF) foraging tasks.

1. Nagele, J., Herz, A. V. M. & Stemmler, M. B. Untethered firing fields and intermittent silences: Why grid-cell discharge is so variable. *Hippocampus* **30**, 367–383 (2020).
2. Dunn, B., Mørreaunet, M. & Roudi, Y. Correlations and Functional Connections in a Population of Grid Cells. *PLOS Comput. Biol.* **11**, e1004052 (2015).
3. Sargolini, F. *et al.* Conjunctive Representation of Position, Direction, and Velocity in Entorhinal Cortex. *Science* **312**, 758–762 (2006).
4. Hinman, J. R., Brandon, M. P., Climer, J. R., Chapman, G. W. & Hasselmo, M. E. Multiple Running Speed Signals in Medial Entorhinal Cortex. *Neuron* **91**, 666–679 (2016).
5. Burak, Y. & Fiete, I. R. Accurate Path Integration in Continuous Attractor Network Models of Grid Cells. *PLoS Comput. Biol.* **5**, e1000291 (2009).
6. Averbeck, B. B., Latham, P. E. & Pouget, A. Neural correlations, population coding and computation. *Nat. Rev. Neurosci.* **7**, 358–366 (2006).
7. Moreno-Bote, R. *et al.* Information-limiting correlations. *Nat. Neurosci.* **17**, 1410–1417 (2014).
8. Kohn, A., Coen-Cagli, R., Kanitscheider, I. & Pouget, A. Correlations and Neuronal Population Information. *Annu. Rev. Neurosci.* **39**, 237–256 (2016).
9. Azeredo Da Silveira, R. & Rieke, F. The Geometry of Information Coding in Correlated Neural Populations. *Annu. Rev. Neurosci.* **44**, 403–424 (2021).
10. Giocomo, L. M., Moser, M.-B. & Moser, E. I. Computational Models of Grid Cells. *Neuron* **71**, 589–603 (2011).

11. Giocomo, L. M., Zilli, E. A., Fransén, E. & Hasselmo, M. E. Temporal Frequency of Subthreshold Oscillations Scales with Entorhinal Grid Cell Field Spacing. *Science* **315**, 1719–1722 (2007).
12. Hardcastle, K., Ganguli, S. & Giocomo, L. M. Environmental Boundaries as an Error Correction Mechanism for Grid Cells. *Neuron* **86**, 827–839 (2015).
13. Bishop, C. M. *Pattern Recognition and Machine Learning*. (Springer New York, 2006).
14. Bush, D., Barry, C. & Burgess, N. What do grid cells contribute to place cell firing? *Trends Neurosci.* **37**, 136–145 (2014).
15. Sorscher, B., Mel, G. C., Ocko, S. A., Giocomo, L. M. & Ganguli, S. A unified theory for the computational and mechanistic origins of grid cells. *Neuron* **111**, 121-137.e13 (2023).
16. Bonnevie, T. *et al.* Grid cells require excitatory drive from the hippocampus. *Nat. Neurosci.* **16**, 309–317 (2013).
17. McClain, K., Tingley, D., Heeger, D. J. & Buzsáki, G. Position–theta-phase model of hippocampal place cell activity applied to quantification of running speed modulation of firing rate. *Proc. Natl. Acad. Sci.* **116**, 27035–27042 (2019).
18. Stachenfeld, K. L., Botvinick, M. M. & Gershman, S. J. The hippocampus as a predictive map. *Nat. Neurosci.* **20**, 1643–1653 (2017).
19. Campbell, M. G., Attinger, A., Ocko, S. A., Ganguli, S. & Giocomo, L. M. Distance-tuned neurons drive specialized path integration calculations in medial entorhinal cortex. *Cell Rep.* **36**, 109669 (2021).

20. Kropff, E., Carmichael, J. E., Moser, M. B. & Moser, E. I. Speed cells in the medial entorhinal cortex. *Nature* **523**, 419–424 (2015).
21. Chaudhuri, R., Gerçek, B., Pandey, B., Peyrache, A. & Fiete, I. The intrinsic attractor manifold and population dynamics of a canonical cognitive circuit across waking and sleep. *Nat. Neurosci.* **22**, 1512–1520 (2019).
22. Higgins, I. *et al.* beta-VAE: Learning Basic Visual Concepts with a Constrained Variational Framework. in *Proceedings of the 5th International Conference on Learning Representations (ICLR)* (2017).
23. Peer, N., Yamin, H. & Cohen, D. Multidimensional encoding of movement and contextual variables by rat globus pallidus neurons during a novel environment exposure task. *iScience* **25**, 105024 (2022).
24. Carpenter, F., Manson, D., Jeffery, K., Burgess, N. & Barry, C. Grid Cells Form a Global Representation of Connected Environments. *Curr. Biol.* **25**, 1176–1182 (2015).
25. Barry, C., Hayman, R., Burgess, N. & Jeffery, K. J. Experience-dependent rescaling of entorhinal grids. *Nat. Neurosci.* **10**, 682–684 (2007).
26. Keinath, A. T., Epstein, R. A. & Balasubramanian, V. Environmental deformations dynamically shift the grid cell spatial metric. *eLife* **7**, e38169 (2018).
27. Barry, C., Ginzberg, L. L., O’Keefe, J. & Burgess, N. Grid cell firing patterns signal environmental novelty by expansion. *Proc. Natl. Acad. Sci.* **109**, 17687–17692 (2012).
28. Schneider, S., Lee, J. H. & Mathis, M. W. Learnable latent embeddings for joint behavioural and neural analysis. *Nature* **617**, 360–368 (2023).

29. Nejatbakhsh, A., Garon, I. & Williams, A. H. Estimating Noise Correlations Across Continuous Conditions With Wishart Processes. *Adv. Neural Inf. Process. Syst.* **36**, 54032–54045 (2023).
30. Kriegeskorte, N. & Wei, X. X. Neural tuning and representational geometry. *Nat. Rev. Neurosci.* **22**, 703–718 (2021).
31. Ye, Z., Li, H., Tian, L. & Zhou, C. Beyond the Delay Neural Dynamics: a Decoding Strategy for Working Memory Error Reduction. *bioRxiv* 2022.06.01.494426 (2022)
doi:10.1101/2022.06.01.494426.

Responses to Reviewer Comments

We thank the Reviewers for their constructive feedback. Please find below our point-by-point responses to the Reviewers' comments. We have revised the manuscript accordingly and highlighted those changes in green.

Reviewer #1, specific comments

***RI.1.** I find the inclusion of GKR-S (added in response to another reviewer) helpful, but I also found its introduction in the manuscript somewhat confusing. The authors state that their main method violates the assumption of normality, but then proceed to use it anyway, later demonstrating that it produces results similar to GKR-S. I think this can be easily fixed, e.g. by introducing GKR-S later and stating there that the main method (GKR) yields comparable results, which also justifies continuing to focus on it as the primary method.*

We thank the reviewer for their suggestions and have made revisions accordingly. We moved all results about GKR-S into a new section: Results from GKR agree with those from a modified method that does not assume data normality (page 11).

***RI.2.** Regarding the IPSG model: it is interesting (if not surprising) that this simplified formulation reproduces many of the key results - up until the results when noise correlations are considered. The model is simplified, as the authors say, but the fact that many results can be reproduced under these simplifications suggests to me that certain complexities in the real data (e.g. nonlinear relationships between running speed and firing rate or non-Poisson firing) are not critical for the core results presented here. This is useful for understanding, as it helps isolate which features must be added to a simplified model to approximate real data, thereby clarifying the functional role of those features. I would thus possibly re-state the impact of this analysis in the paper.*

We agree with the reviewer that the IPSG results merit deeper interpretation. Accordingly, we have added a paragraph on page 12 that emphasizes the key features explaining the speed-modulation effects. That paragraph is also pasted below:

Intuitively, in IPSG, increasing speed amplifies each cell's firing without altering its spatial tuning. Because manifold size scales roughly with firing rate, it too grows with speed (see Methods, Figure 3E, SI Figure 12). Poisson-neuron assumption implies that noise (the standard deviation) scales as the square root of the firing rate, so noise increases more slowly than firing rate. As a result, each cell's signal-to-noise ratio—and hence its Fisher information—rises with speed under the independent-firing assumption (see Methods). More loosely, IPSG suggests that faster running increases grid-cell firing (without disrupting the rate map too much) more than noise and the effects of noise correlations are negligible, explaining the observed speed modulations on the manifold (Figure 3E, 4C, and 5B). Whether this picture holds in real grid-cell data remains unclear. We leave more precise theoretically modeling and validations for future work, while this paper focuses more on data-driven descriptive analysis.

R1.3. Alongside this, the IPSG model, as the authors note, cannot capture the effects of noise correlations. I wonder if therefore, it can be viewed as a null model for those analyses, essentially replacing the IFGC model, which would simplify presentation.

Indeed, IPSG assumes neuronal noise independence; however, we do not consider it an appropriate null model for real neural data. Besides firing independence, IPSG relies on other strong assumptions, such as that running speed acts as a gain factor on grid-cell firing and that spatial tuning can be modeled as a sum of cosine functions. Even if we find differences between IPSG results and the real data, it is difficult to determine which assumption(s) underlie these discrepancies.

On the other hand, the IFGC model makes minimal modifications to the dataset. In Figure 6E, IFGC results were obtained by only shuffling data across “trials” (i.e., within very localized ranges of speed and x–y position; see Methods: Independent Firing Grid Cells, page 31–32) to disrupt noise correlations without changing, for example, the mean firing or total noise etc. It is therefore easier to draw conclusions from comparisons to IFGC.

Reviewer #3, specific comments

R3.1. When the authors discussed this part (R3.1 What is the implication of the results on the underlying model of grid cells?), I think the authors need to consider:

1. Normally, there are three types of models of grid cells: CAN (continuous attractor network, rate-based), OI (Oscillatory-interference) and self-organising (learning models).

We agree with the reviewer that the self-organizing model differs from a continuous attractor network (CAN). For simplicity, however, we decided to group both under “rate-based models,” since both rely on firing rate and network interactions. We have revised the discussion to explicitly include self-organising models (page 15), also see below:

We analyzed the population-level properties of grid cell activities, but what are the implications of these findings for individual grid cells? There are two major models of grid cells: the rate-based model and the oscillatory-interference model. First, in some rate-based models (continuous attractor networks), running speed serves as an input to the grid-cell network. Faster speeds may elevate firing rates, which can sharpen the spatial rate map, enhance the signal-to-noise ratio, and thus boost population Fisher information (Figure 5). Although, some other rate-based models contain normalization mechanisms implying the opposite—that the population mean firing rate should not change with speed (self-organising models).

R3.2. Kropff et al. 2021 mentioned that theta frequency is controlled by acceleration (not speed). How to reconcile with this finding?

This is a very interesting question. Kropff et al. (2021) demonstrated that LFP theta frequency is controlled by acceleration rather than speed; however, they also reported a mild modulation of grid cell intrinsic firing frequency by running speed (Figures S3–S4 in Kropff et al. 2021)¹. Further, their study also does not rule out other possible speed-dependent effects on grid cells. Therefore, we think speed modulation of grid cell spatial representation remains possible.

We thank the reviewer for this insight and have added a brief discussion of Kropff et al. (2021) in Ln. 384 (page 15).

Reviewer #4 (Remarks to the Author):

I want to thank the authors for their detailed answers. I have no more comments or questions.

We thank the reviewer for their constructive comments, which have helped improve the manuscript.

References

1. Kropff, E., Carmichael, J. E., Moser, E. I. & Moser, M.-B. Frequency of theta rhythm is controlled by acceleration, but not speed, in running rats. *Neuron* **109**, 1029-1039.e8 (2021).